# XCP1 cleaves Pathogenesis-related protein 1 into CAPE9 for systemic immunity in *Arabidopsis*

Ying-Lan Chen [1,2], Fan-Wei Lin [1], Kai-Tan Cheng [1], Chi-Hsin Chang [1,3,4], Sheng-Chi Hung [1,5], Thomas Efferth[6] & Yet-Ran Chen [1,3,4,5] ✉

Proteolytic activation of cytokines regulates immunity in diverse organisms. In animals, cysteine-dependent aspartate-specific proteases (caspases) play central roles in cytokine maturation. Although the proteolytic production of peptide cytokines is also essential for plant immunity, evidence for cysteine-dependent aspartate-specific proteases in regulating plant immunity is still limited. In this study, we found that the C-terminal proteolytic processing of a caspase-like substrate motif "CNYD" within Pathogenesis-related protein 1 (PR1) generates an immunomodulatory cytokine (CAPE9) in *Arabidopsis*. Salicylic acid enhances CNYD-targeted protease activity and the proteolytic release of CAPE9 from PR1 in *Arabidopsis*. This process involves a protease exhibiting caspase-like enzyme activity, identified as Xylem cysteine peptidase 1 (XCP1). XCP1 exhibits a calcium-modulated pH-activity profile and a comparable activity to human caspases. XCP1 is required to induce systemic immunity triggered by pathogen-associated molecular patterns. This work reveals XCP1 as a key protease for plant immunity, which produces the cytokine CAPE9 from the canonical salicylic acid signaling marker PR1 to activate systemic immunity.

Proteases, which hydrolyze proteins into shorter proteins, peptides, or amino acids, are involved not only in protein turnover but also in the regulation of diverse physiological events[1]. For instance, proteases regulate immunity by generating peptides from host or pathogen precursors; these peptides activate and orchestrate defense responses to defeat biological threats[2]. In the animal kingdom, members of the cysteine-dependent aspartate-specific protease (caspase) family serve as central mediators in the initiation and execution of apoptosis, as well as the activation of inflammation via proteolytic maturation of cytokines[3]. Caspases belong to a cysteine protease family with high specificity; they recognize a motif of at least four amino acids that ends in and cleaves immediately after

aspartate (xxxD↓x)[4]. Although plants are expected to have caspases or their functional analogs[5], no sequence homolog of caspases has been directly identified[6], and no plant protease has been found to cleave a precursor after aspartate to produce a mature cytokine directly. Currently, the only known functional analogs of animal caspases in plants are metacaspases (a type of cysteine protease that cleaves lysine/arginine), phytaspases (a type of serine proteases that cleaves aspartate), vacuolar processing enzymes (a type of cysteine protease that exhibits caspase-1 activity) and 20 S proteasome subunit PBA1 (a type of threonine protease that exhibits caspase-3 activity)[7–10]. Although caspase-like activity has been reported in plants[9–11], and plant immune responses can be initiated by the

[1]Agricultural Biotechnology Research Center, Academia Sinica, Taipei 115, Taiwan. [2]Department of Biotechnology and Bioindustry Sciences, College of Bioscience and Biotechnology, National Cheng Kung University, Tainan 701, Taiwan. [3]Molecular and Biological Agricultural Sciences Program, Taiwan International Graduate Program, Academia Sinica, Taipei 115, Taiwan. [4]Taiwan Graduate Institute of Biotechnology, National Chung-Hsing University, Taichung 402, Taiwan. [5]Institute of Biotechnology, National Taiwan University, Taipei 106, Taiwan. [6]Department of Pharmaceutical Biology, Institute of Pharmaceutical and Biomedical Sciences, Johannes Gutenberg University, Mainz, Germany. ✉e-mail: yetran@gate.sinica.edu.tw

maturation of plant immunomodulatory peptide cytokines[12], most of the underlying enzymes are not clear.

Among the known immunomodulatory peptides in plants, a tomato wound-induced peptide is produced by a cleavage event that is after a caspase-like substrate motif, "CNYD↓", within the proprotein "Pathogenesis-related protein 1" (PR1)[13]. PR1 belongs to the cysteine-rich secretory protein, antigen 5, and pathogenesis-related 1 protein (CAP) superfamily[14], thus, the first identified mature peptide was named CAP-derived peptide 1 (CAPE1)[13]. Tomato CAPE1 (SlCAPE1) induces antipathogen and minor anti-herbivore responses without significantly triggering programmed cell death. SlCAPE1 and the CNYD domain positioned N-terminal to SlCAPE1 are highly conserved within PR1 across diverse plant species. PR1 is the most common marker for salicylic acid (SA)-regulated plant immunity and its secretion is critical for the activation of systemic acquired resistance (SAR) in *Arabidopsis*[13]. Nevertheless, how PR1 regulates SAR is poorly understood. In *Arabidopsis*, CAPE9 is a putative CAPE that is also derived from PR1, and the treatment with synthetic *Arabidopsis* CAPE9 (AtCAPE9) induces antipathogen activity in *Arabidopsis*[13]. However, endogenous AtCAPE9 has not yet been detected, and whether PR1 is processed into AtCAPE9 to regulate *Arabidopsis* systemic immunity still remains unknown. In this study, we demonstrate that the production of AtCAPE9 from PR1 is essential for *Arabidopsis* SAR and involves an enzyme specific for CAPE production, which is Xylem cysteine peptidase 1 (XCP1). We show that the XCP1 recognizes and cleaves the end peptide bond of the caspase-like motif CNYD in PR1 to release AtCAPE9 and that the presence of XCP1 is essential for eliciting systemic plant immunity. Our findings highlight the critical role of PR1 as pro-cytokine and the significance of a caspase-like enzyme in eliciting plant systemic immunity.

## Results

### AtCAPE9 is generated from aspartate-specific proteolysis of PR1 and enhances immunity in *Arabidopsis*

To decipher the function of AtCAPE9 in regulating *Arabidopsis* immunity, we treated two groups of *Arabidopsis* plants with water or an aqueous solution of synthetic AtCAPE9 (PRGNYVNEKPY). Compared to the plants treated with water, the AtCAPE9-treated plants displayed an increased level of SA (2.1-fold) (Fig. 1a). Similar to the AtCAPE9 treatment reduced the pathogen infection in adult leaves[13], the AtCAPE9-treated seedling also showed a reduced infection upon inoculation with *Pseudomonas syringae* pv. *tomato* DC3000 (*Pst* DC3000) in contrast to water-treated control (72.7-fold) (Fig. 1a). We performed LC−MS/MS to identify endogenous AtCAPE9 in SA-treated leaves, whose MS/MS spectrum closely matched that of synthetic AtCAPE9 with a Pearson correlation coefficient of 0.95 (Fig. 1b, Supplementary Data 1). Using an untargeted peptidomics approach that we had proposed previously[13], 615 nonredundant peptides in SA-treated leaves were identified (Supplementary Data 2) and found that AtCAPE9 was the only peptide derived from the CAPE family (Supplementary Fig. 1). Therefore, AtCAPE9 is the primary peptide cytokine in the CAPE family in SA treated plant that can regulate SA levels and enhances resistance to bacterial pathogens in *Arabidopsis*.

To investigate PR1 processing depending on the aspartate of the CNYD motif that is N-terminal to the AtCAPE9 sequence within *Arabidopsis* PR1, we individually overexpressed different sequences of PR1 fused to an enhanced yellow fluorescent protein (eYFP) in *Arabidopsis*. Specifically, we examined native (N: CNYDP) or alanine-substituted versions (D150A: CNYAP or P151A: CNYDA) of PR1-eYFP (Fig. 1c). We detected a ~44.7 kDa protein with all three constructs, corresponding to intact/uncleaved form of PR1-eYFP. In contrast, a ~27.0 kDa protein close to the size of the putative AtCAPE9-eYFP cleavage product was

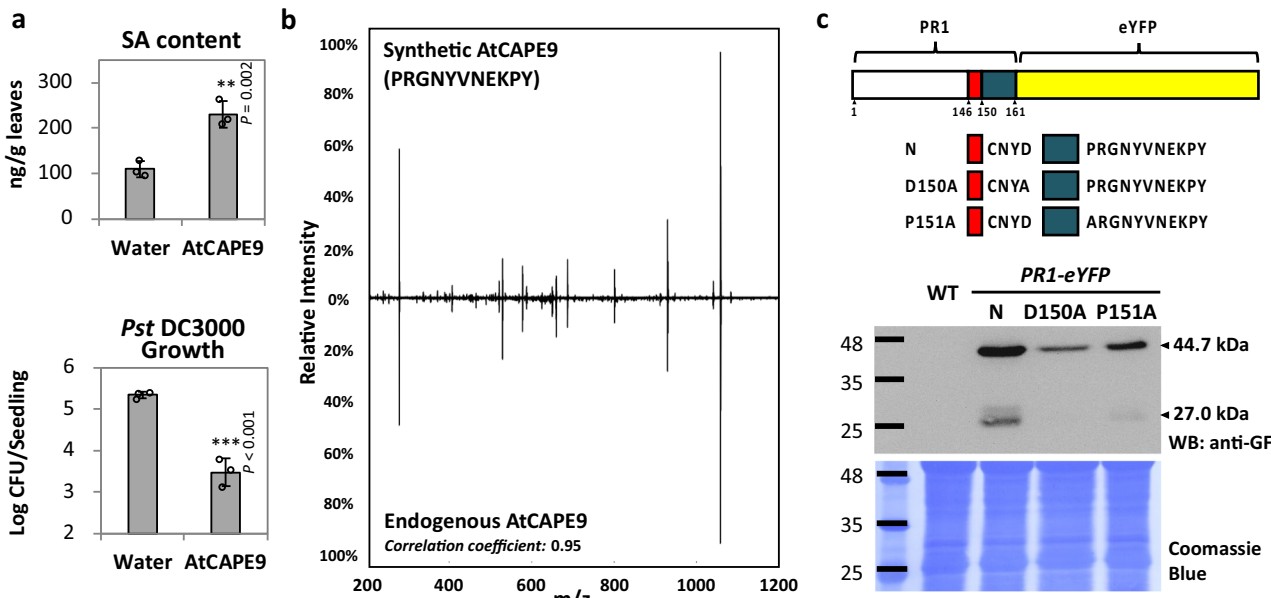

**Fig. 1 | Bioactivity and endogenous production of AtCAPE9 through proteolysis of PR1 in *Arabidopsis*. a** The SA contents and the growth of *Pst* DC3000 in *Arabidopsis* treated with water or AtCAPE9. The SA concentration in the plants was quantified by LC−MS/MS with spiked deuterium-labeled SA standard. Log colony-forming units (Log CFU) of *Pst* DC3000 were measured after 7 days of inoculation. Values are means ± SD of three biological replicates. Each replicate was obtained from the pooling of three plants. *P* values were calculated by one-tailed unpaired *t*-test (**, *P* < 0.01; ***, *P* < 0.001). **b** MS/MS spectra of the endogenous and synthetic AtCAPE9. The endogenous AtCAPE9 was identified from the SA-treated *Arabidopsis* leaves. The peak list for each spectrum and the similarity between the spectra were

determined by Pearson correlation coefficient calculated in Supplementary Data 1. **c** Immunoblot of the native or alanine-substituted (N, D150A or P151A) PR1 fused to an enhanced yellow fluorescent protein (eYFP) overexpressed in *Arabidopsis*. A schema representing three constructs for each transgenic line is illustrated at the top of the panel. The sizes of intact/uncleaved PR1-eYFP (N, D150A and P151A) proteins and the AtCAPE9-eYFP fragment were estimated to be ~44.7 and ~27.0 kDa, respectively, detected by western blotting with anti-GFP antibody. Coomassie Blue staining shows total protein loaded. Experiments were repeated three times with similar results.

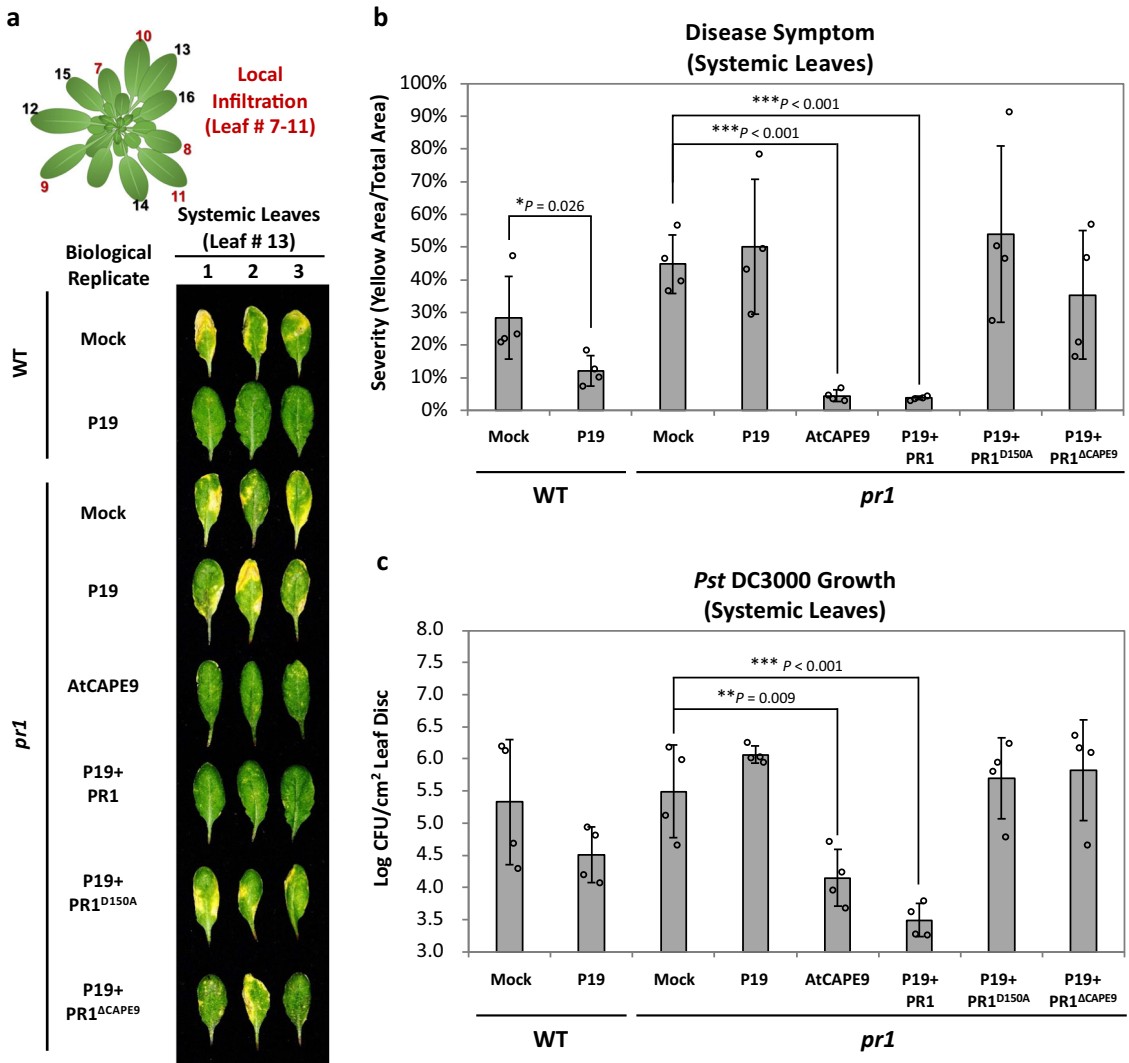

**Fig. 2 | Systemic immunity induced by locally transforming different forms of PR1 to *pr1*. a** The disease symptom of the *Pst* DC3000-infected systemic leaves of wild-type (WT) and *pr1* mutant pre-infiltrated local leaves with different agents. Agents including the *Agrobacterium* infiltration buffer (Mock), or *Agrobacterium* carrying P19 plasmid (P19), buffer with AtCAPE9 (AtCAPE9), or *Agrobacterium* carrying P19 plasmid together with *Agrobacterium* carrying native *PR1* (P19 + PR1), alanine-substituted *PR1* (P19 + PR1^D150A), or CAPE9-truncated *PR1* (P19 + PR1^ΔCAPE9) driven by 35S promotor. The local pre-infiltrations were performed on leaves # 7–11, and other leaves without infiltration were used as systemic leaves. After 4 days of local infiltration, both local and systemic leaves were inoculated with *Pst* DC3000 for 7 days, and collected systemic leaves were for disease symptom assessment. **b** Levels of disease severity were measured by the percentage of yellow area in the total area for the corresponding photographs using the PIDIQ software[52]. **c** Log colony-forming units (Log CFU) of *Pst* DC3000 were measured after 7 days of inoculation. **b**, **c** Values are means ± SD of four biological replicates. Each replicate was obtained from the systemic leaves of three individual plants. *P* values were calculated by one-tailed unpaired *t*-test (*, *P* < 0.05; **, *P* < 0.01; ***, *P* < 0.001).

produced from the native and P151A mutant of PR1-eYFP but not from the D150A mutant. These data suggest that the aspartate in the potential caspase-like substrate domain CNYD is important for PR1 cleavage for CAPE production.

To investigate whether the CNYD and the CAPE motifs in PR1 are involved in systemic immunity, we overexpressed different forms of PR1 in the *pr1* mutant of *Arabidopsis* locally and examined the disease resistance of systemic leaves against *Pst* DC3000 (Fig. 2). Here, we generated a *pr1* mutant using CRISPR/Cas9 (Supplementary Fig. 2a–c) and then performed local infiltration in the selected leaves of *pr1* with buffer (mock), *Agrobacterium* carrying *P19* plasmid (P19), buffer with AtCAPE9 peptide solution (AtCAPE9), or P19 individually mixed with *Agrobacterium* carrying native *PR1* (P19 + PR1), alanine-substituted (D150A) *PR1* (P19 + PR1^D150A), or AtCAPE9-truncated *PR1* (P19 + PR1^ΔCAPE9) plasmids driven by 35S promoter (Supplementary Fig. 2d, e). Relative to the mock treatment, the local infiltration of the

*Agrobacterium* carrying *P19* plasmid only (P19) can trigger systemic immunity to reduce the disease severity of the systemic leaves infected by *Pst* DC3000 in wild-type *Arabidopsis* (WT), but not in *pr1* (Fig. 2b). In *pr1*, local treatment of AtCAPE9 or transient expression of P19 plus native PR1 were both able to significantly reduce the disease severity (Fig. 2b) and pathogen population (Fig. 2c) in the systemic leaves in comparison to the local mock treatment, or transient expression of P19, P19 plus PR1^D150A or P19 plus PR1^ΔCAPE9. This study indicated that both CNYD and CAPE domains for proteolytic production of AtCAPE9 from PR1 are crucial for eliciting *Arabidopsis* systemic immunity.

## SA increases AtCAPE9 level, PR1-eYFP cleavage, and CNYD-targeted protease (CNYDase) activity in *Arabidopsis*

We hypothesized that AtCAPE9, like PR1, is involved in SA-triggered immunity in *Arabidopsis*. To determine if SA treatment induces the

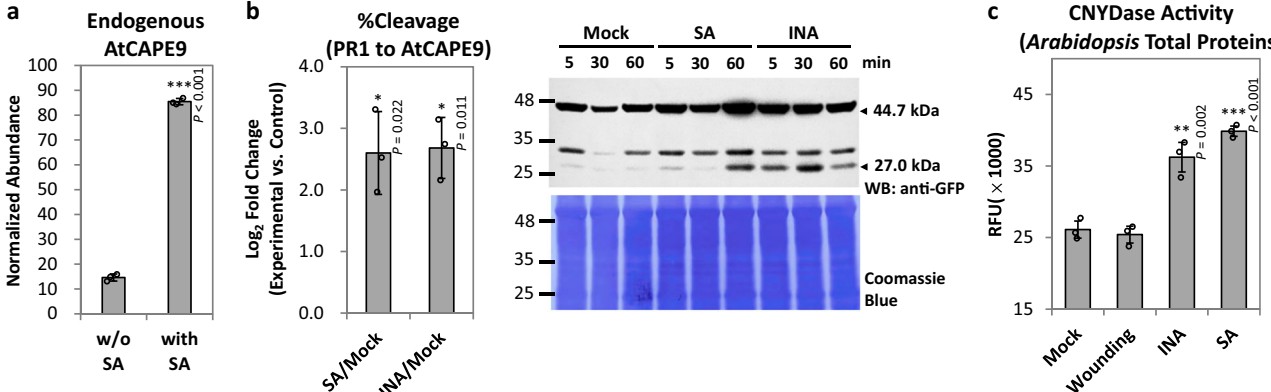

**Fig. 3 | Induction of AtCAPE9 production, PR1-eYFP cleavage, and CNYDase activity by SA or SA functional analog INA. a** The abundance of endogenous AtCAPE9 in *Arabidopsis* leaves with or without SA treatment. The abundance of AtCAPE9 was quantified and normalized to the abundance of a selected peptide signal from spiked tryptic β-casein using LC−MS/MS. Values are means ± SD of three technical replicates obtained from the analysis of a peptide mixture extracted from 50 g leaf tissues derived from a pool of 50 plants. **b** Left panel: The cleavage percentage of PR1-eYFP for AtCAPE9-eYFP production (%Cleavage) in transgenic *PR1-eYFP Arabidopsis* seedlings treated by SA or INA for 60 min. The %Cleavage was calculated from the band intensity of AtCAPE9-eYFP divided by the sum of PR1-eYFP and AtCAPE9-eYFP band intensities detected by immunoblot using anti-GFP (Supplementary Fig. 3). Values are means ± SD of three biological replicates. Each replicate was obtained from the pooling of six plants. Right panel: Immunoblot of PR1-eYFP and AtCAPE9-eYFP fragment in transgenic *PR1-eYFP Arabidopsis* seedlings treated with Mock, INA, or SA for 5, 30, and 60 min. The sizes of the intact/uncleaved PR1-eYFP and the putative AtCAPE9-eYFP fragment were estimated to be ~44.7 and ~27.0 kDa, respectively, detected by immunoblot using anti-GFP. Coomassie Blue staining shows total protein loaded. **c** CNYDase activity of the protein extract in wild-type (WT) *Arabidopsis* plants. The plants were sprayed with $MgSO_4$ (Mock), wounded, or treated with $MgSO_4$ plus INA or SA. Lysate was collected 24 h later. The proteolytic activity assay was performed by incubating 50 μg protein extract with 25 μM Ac-CNYD-AMC substrate. RFU of the cleaved fluorophore was measured after 10 h incubation with the substrate. Values are means ± SD of three biological replicates. Each replicate was obtained from pooled tissues of three plants. *P* values were calculated by one-tailed unpaired *t*-test (*, $P < 0.05$; **, $P < 0.01$; ***, $P < 0.001$).

production of AtCAPE9, we used LC−MS/MS to quantify the level of endogenous AtCAPE9 in plants with or without SA treatment. Briefly, LC−MS was operated in targeted MS/MS mode by selecting the precursor and specific fragment ions of AtCAPE9. We found that the level of AtCAPE9 was ~6-fold higher in SA-treated plants compared to the untreated controls (Fig. 3a). Moreover, the cleavage of PR1-eYFP to produce AtCAPE9-eYFP was significantly enhanced in transgenic plants treated with SA or the SA functional analog 2,6-dichloroisonicotinic acid (INA) compared to mock-treated controls (Fig. 3b; Supplementary Fig. 3).

To monitor plant CNYDase activity, we synthesized an N-acetyl CNYD peptide tagged with 7-amino-4-methylcoumarin (Ac-CNYD-AMC) as fluorogenic protease substrate (Supplementary Fig. 4) and incubated it with plant protein extracts. Substrate cleavage was elevated in extracts from SA-treated and INA-treated plants, compared to extracts from mock-treated or wounded plants (Fig. 3c), suggesting that CNYDase activity increases upon SA and INA treatment. Together, these data suggest that SA treatment enhances total CNYDase activity to generate increased levels of AtCAPE9 from PR1 in *Arabidopsis*.

### Identification of a putative cysteine protease targeting the CNYD motif in *Arabidopsis*

Next, we used the CNYDase assay to further examine whether the conserved CNYD motif in PR1 is essential for the proteolytic release of AtCAPE9. Indeed, we observed substantially reduced cleavage of fluorogenic substrates with a disrupted CNYD motif (Ac-CNAD-AMC, Ac-ANAD-AMC) compared to the canonical Ac-CNYD-AMC substrate (Fig. 4a). In addition, cleavage of the Ac-CNYD-AMC substrate was significantly enhanced by $CaCl_2$, but inhibited by $ZnCl_2$ and the metal chelator EDTA, as compared to the control (Fig. 4b). The $Ca^{2+}$-enhanced CNYDase activity was strongly suppressed by a general cysteine protease inhibitor (E-64) or a biotinylated aldehyde tetrapeptide CNYD protease inhibitor (biotin-CNYD-CHO; Supplementary Fig. 4, Fig. 4c). The suppression by E-64 suggested that the CNYDase is a cysteine protease that catalyzes a cysteine-dependent proteolytic reaction.

The biotin−CNYD−CHO inhibitor was designed to probe and covalently modify the cysteine in the active site of CNYDases, and displayed dose-dependent inhibition of CNYDase activity with the Ac−CNYD−AMC substrate (Fig. 4d). The aldehyde group of the inhibitor is able to form covalent hemithioacetal adduct mimicking the tetrahedral transition states of the cysteine protease reaction in catalytic center[15]. We incubated biotin−CNYD−CHO with wild-type (WT) *Arabidopsis* extract and detected biotin−CNYD−CHO-labeled proteins with streptavidin−HRP by western blotting. Intriguingly, we observed prominent labeling of a potential 35 kDa CNYDase and two minor protein bands at ~65 and ~24 kDa (Fig. 4e). Together, our results so far point to a protease that is mainly 35 kDa, $Ca^{2+}$-activated cysteine-dependent and aspartate-specific CNYDase in *Arabidopsis*.

### XCP1 is a CNYDase enzyme specific for CAPE production from PR1

To discover the enzyme specific for CAPE production (designated ESCAPE), we investigated two cysteine protease families in *Arabidopsis*, the papain-like cysteine proteases (PLCPs) and metacaspases (MCs), which have caspase-like functions in regulating plant immunity and programmed cell death[16,17]. The MC family members are not aspartate-specific and therefore were excluded as candidates. Among the 31 *Arabidopsis* PLCPs, we identified ESCAPE candidates that had: (1) a molecular weight of 34.00–37.00 kDa (untruncated, pro-form) and 23.00–25.00 kDa (truncated, processed form) (Supplementary Table 1), and (2) an expression pattern that correlates with *PR1* expression during leaf development[18] (Supplementary Fig. 5). The only PLCP, Xylem cysteine peptidase 1 (XCP1; At4g35350), was fitting to both criteria (Supplementary Fig. 6a). In addition, the ~35 kDa untruncated form of XCP1 has been shown to be significantly more abundant than the ~23 kDa truncated form in *35S:XCP1* plants[19]. However, Xylem cysteine peptidase 2 (XCP2; At1g20850) also localizes in the xylem and its putative truncation and untruncated form both have a similar size as XCP1. Therefore, we further tested 9 PLCPs mutants, including *xcp1* and *xcp2*, and found out that only *xcp1* mutant has significant lower lysate CNYDase activity compared to the WT

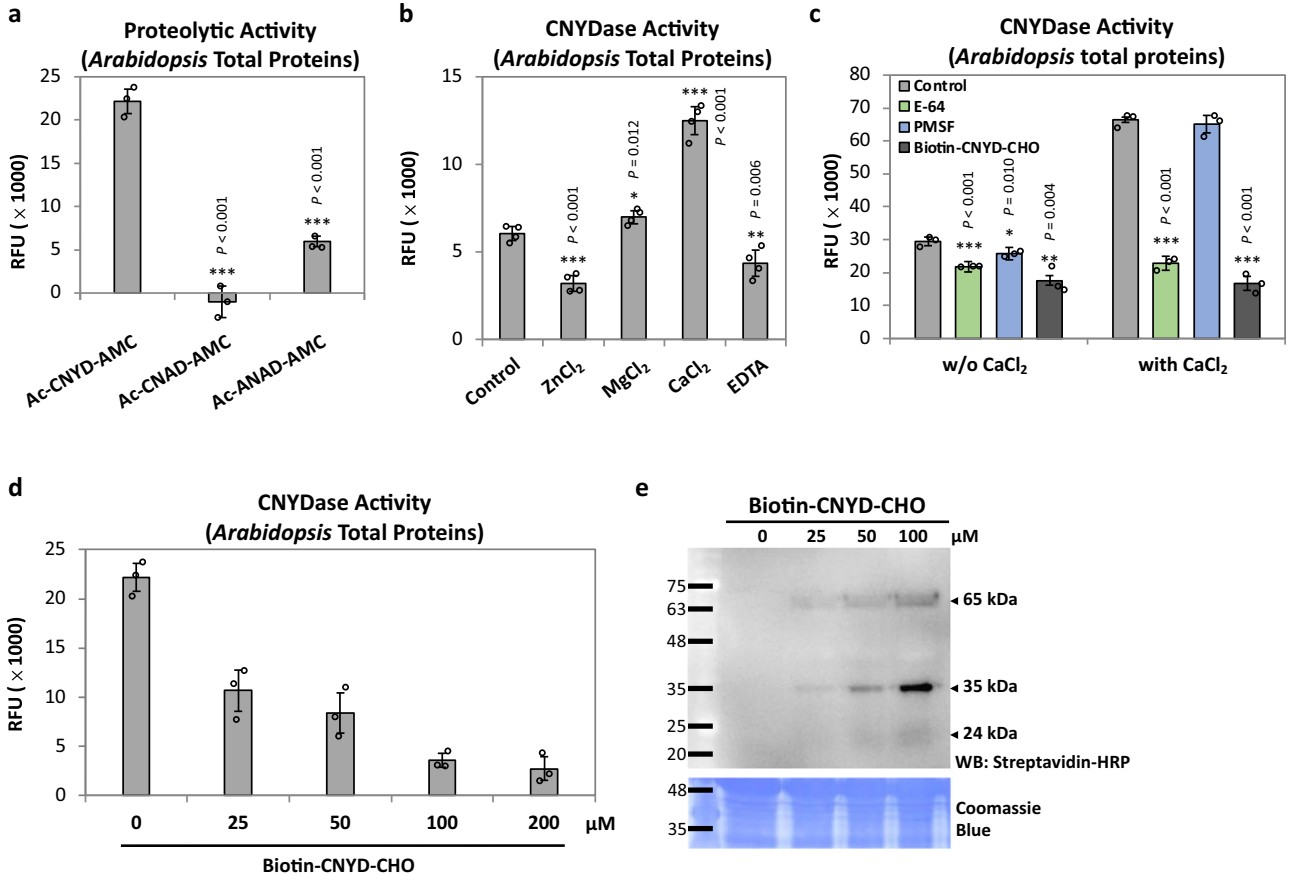

**Fig. 4 | Proteolytic specificity of CNYD motif and the properties of CNYD-targeted protease in *Arabidopsis*. a** Examination of the proteolytic activity of *Arabidopsis* protein extract using three fluorogenic protease substrates (CNYD, CNAD, and ANAD). **b** CNYDase activity of *Arabidopsis* protein extract supplemented with buffer only (control), $ZnCl_2$, $MgCl_2$ or $CaCl_2$, or EDTA. **c** CNYDase activity of *Arabidopsis* protein extract supplemented with buffer only (control), PMSF, E-64, or biotin−CNYD−CHO under the condition with or without $CaCl_2$ for 1 h prior to substrate incubation. **d** CNYDase activity and **e** gel blot of biotinylated proteins from *Arabidopsis* protein extract incubated with different concentrations of biotin−CNYD−CHO. Biotinylated proteins were detected by western blotting with streptavidin−HRP, and Coomassie Blue staining shows total protein loaded. **a**, **b** Relative fluorescent units (RFU) of the cleaved fluorophores were measured after 5 h incubation with the substrate. **c**, **d** RFU were measured after 10 h incubation with the substrate. Each assay was performed by incubating 50 μg protein extract from untreated *Arabidopsis* with 25 μM substrate. Values are means ± SD of three biological replicates. Each replicate was obtained from pooled tissues of three plants. *P* values were calculated by one-tailed unpaired *t*-test (*, $P < 0.05$; **, $P < 0.01$; ***, $P < 0.001$).

(Supplementary Fig. 6b). The *xcp1* mutant showing reduced lysate CNYDase activity (SALK_084789) was determined to be a homozygous T-DNA insertion mutant (Supplementary Fig. 6d). Unlike WT, biotin−CNYD−CHO did not significantly label any other proteins in the *xcp1* mutant extract (Fig. 5a), indicating that XCP1 is the major protease contributing CNYDase activity. Together these data suggest that XCP1 is a primary enzyme in plants that directly recognizes the CNYD motif and catalyzes a proteolytic reaction on the aspartate.

To understand whether the PR1 cleavage and AtCAPE9 production are reduced in *xcp1* mutant, we generated a crossed line with over-expressed *PR1-eYFP* plasmid in the *xcp1* mutant (*xcp1* × *PR1-eYFP*) and examined the AtCAPE9 production in the *xcp1* mutant. The *XCP1* and *PR1-eYFP* gene expression were confirmed by RT-PCR between WT, *PR1-eYFP*, and *xcp1* × *PR1-eYFP* plants (Supplementary Fig. 7a). The PR1-eYFP cleavage to produce AtCAPE9-eYFP was significantly reduced in *xcp1* × *PR1-eYFP* compared to *PR1-eYFP* plants (Supplementary Fig. 7b). Further, we found that the endogenous AtCAPE9 level in WT was significantly reduced in *xcp1* under SA treatment (Supplementary Fig. 7c). The results suggest that the *XCP1* expression affects the PR1 cleavage and AtCAPE9 production in *Arabidopsis*.

To further investigate the activity of XCP1, an XCP1-His fusion protein was expressed in *Nicotiana benthamiana* (*N. benthamiana*) leaves and then purified by Ni-NTA affinity chromatography. The

expected molecular weights of the untruncated and truncated XCP1-His proteins are 37.6 kDa and 24.6 kDa, respectively (Supplementary Fig. 8a), and we detected purified His-tagged proteins of ~45 kDa, 32−37 kDa, and ~25 kDa by immunoblotting (Supplementary Fig. 8b). The 45 kDa protein has been suggested to be a preprotein form of XCP1, which can be observed when the protein is overexpressed[19]. Among the different forms of the XCP1-His protein that were expressed and purified from *N. benthamiana*, the 34−36 kDa and 25 kDa forms could be labeled by biotin−CNYD−CHO (Supplementary Fig. 8b), suggesting that these forms are active CNYDases. Purified XCP1-His displayed proteolytic activity with the Ac-CNYD-AMC substrate but not with the Ac−CNAD−AMC nor Ac−ANAD−AMC substrates, and this activity was efficiently inhibited by co-incubation with biotin−CNYD−CHO (Fig. 5b). Moreover, the CNYDase activity of purified XCP1-His was enhanced by $CaCl_2$ and inhibited by $ZnCl_2$ and E-64, and slightly reduced by EDTA as compared to the mock control (Fig. 5c), similar to our observations with WT *Arabidopsis* lysate. We found that purified XCP1-His exhibited the highest CNYDase activity at pH 6.0 in the presence of excess $CaCl_2$ and at pH 5.0 in the absence of excess $CaCl_2$ (Fig. 5d).

In addition, purified XCP1-His (1 μg) was substantially more active at 22 °C as compared to 32 °C and 37 °C (Supplementary Fig. 9). The triggering of the AtCAPE9-eYFP production by INA in

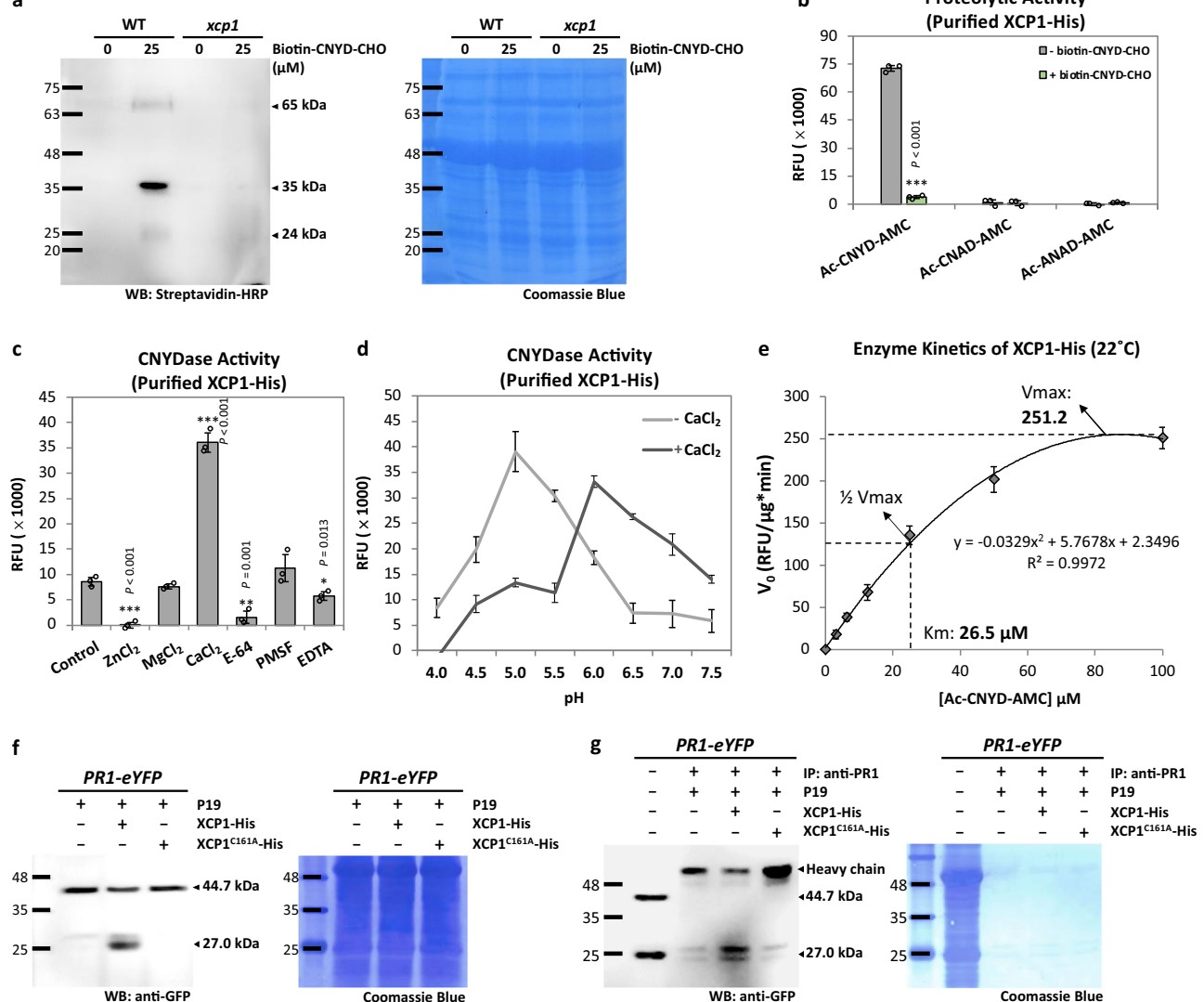

**Fig. 5 | Characterization of the enzyme specific for CAPE production (ESCAPE) by CNYDase activity and interaction with PR1-eYFP. a** Gel blot of biotinylated proteins from *Arabidopsis* wild-type (WT) and *xcp1* extracts with or without the addition of biotin−CNYD−CHO. The biotinylated proteins were detected with streptavidin−HRP. Coomassie Blue staining shows total protein loaded.
**b** Proteolytic activity of purified XCP1-His on three fluorogenic protease substrates (CNYD, CNAD, and ANAD) with or without adding biotin−CNYD−CHO. The proteolytic activity of purified XCP1-His was measured by RFU of the cleaved fluorophore after 10 h incubation with the substrate. **c** CNYDase activity of purified XCP1-His supplemented with $ZnCl_2$, $MgCl_2$, $CaCl_2$, E-64, PMSF, or EDTA for 1 h before substrate incubation. RFU of the cleaved fluorophore was measured after 10 h incubation with the substrate. **d** The CNYDase activity-pH profile of purified XCP1-His, with or without adding $CaCl_2$. RFU of the cleaved fluorophore was measured after 10 h incubation with the substrate. **e** The enzyme kinetics of purified XCP1-His for CNYD substrate. The $V_{max}$ and $K_m$ of XCP1-His proteolytic activity were determined by 10 h incubation of 0.2 µg purified XCP1-His with different concentrations of CNYD substrate at 22 °C. **f** Immunoblots of the protein extract

from *Arabidopsis PR1-eYFP* transgenic plants with the addition of the Ni-NTA purified proteins from the *Nicotiana benthamiana* overexpressing *P19*, *P19* plus *XCP1-His* or *P19* plus *XCP1^C161A^-His* gene. PR1-eYFP was detected with anti-GFP antibodies. Coomassie Blue staining shows total protein loaded. **g** Immunoblot of the immobilized PR1-eYFP with the addition of the Ni-NTA purified proteins from the *Nicotiana benthamiana* overexpressing *P19*, *P19* plus *XCP1-His* or *P19* plus *XCP1^C161A^-His* gene. The protein extract from *Arabidopsis PR1-eYFP* transgenic plants was immunoprecipitated with anti-PR1 (IP: anti-PR1) before incubation with XCP1-His. **a**, **f**, **g** Experiments were repeated three times with similar results. **b**−**d** Each proteolytic activity assay for purified XCP1-His was performed by incubating 1 µg protein with 25 µM substrate. **f**, **g** the sizes of intact PR1-eYFP and AtCAPE9-eYFP were estimated to be ~44.7 and ~27.0 kDa, respectively, detected by the anti-GFP. The Coomassie Blue staining shows total protein loaded. **b**−**e** Values are means ± SD of $n = 3$ independent experiments. Each experimental result was obtained by using five *Nicotiana benthamiana* plants overexpressing XCP1-His transiently. *P* values were calculated by one-tailed unpaired *t*-test (*, $P < 0.05$; **, $P < 0.01$; ***, $P < 0.001$).

*PR1-eYFP* transgenic plant was also higher at 22 °C as compared to 32 °C (Supplementary Fig. 10). At 22 °C, purified XCP1-His (0.2 µg) cleaved Ac-CNYD-AMC with a $K_m$ of ~26.5 µM and $V_{max}$ of ~251.2 RFU/min (Fig. 5e). To determine whether the cysteine (C161) in the putative active site of XCP1[20,21] is important for CNYDase activity, we expressed and purified an alanine-substituted (C161A) XCP1-His mutant (Supplementary Fig. 11). Indeed, no CNYDase activity was detected from XCP1^C161A^-His (Supplementary Fig. 11c). Together,

these data suggest that XCP1 is the cysteine protease specific for CNYDase activity in *Arabidopsis*.

Importantly, we observed increased production of AtCAPE9-eYFP (~27.0 kDa) from PR1-eYFP in WT *Arabidopsis* lysate supplemented with Ni-NTA-purified proteins from *N. benthamiana* expressing P19 plus XCP1-His versus P19 alone or expression with XCP1^C161A^-His (Fig. 5f). To determine if XCP1-His can process PR1-eYFP directly, we immobilized PR1-eYFP on protein A/G beads with anti-PR1 antibody,

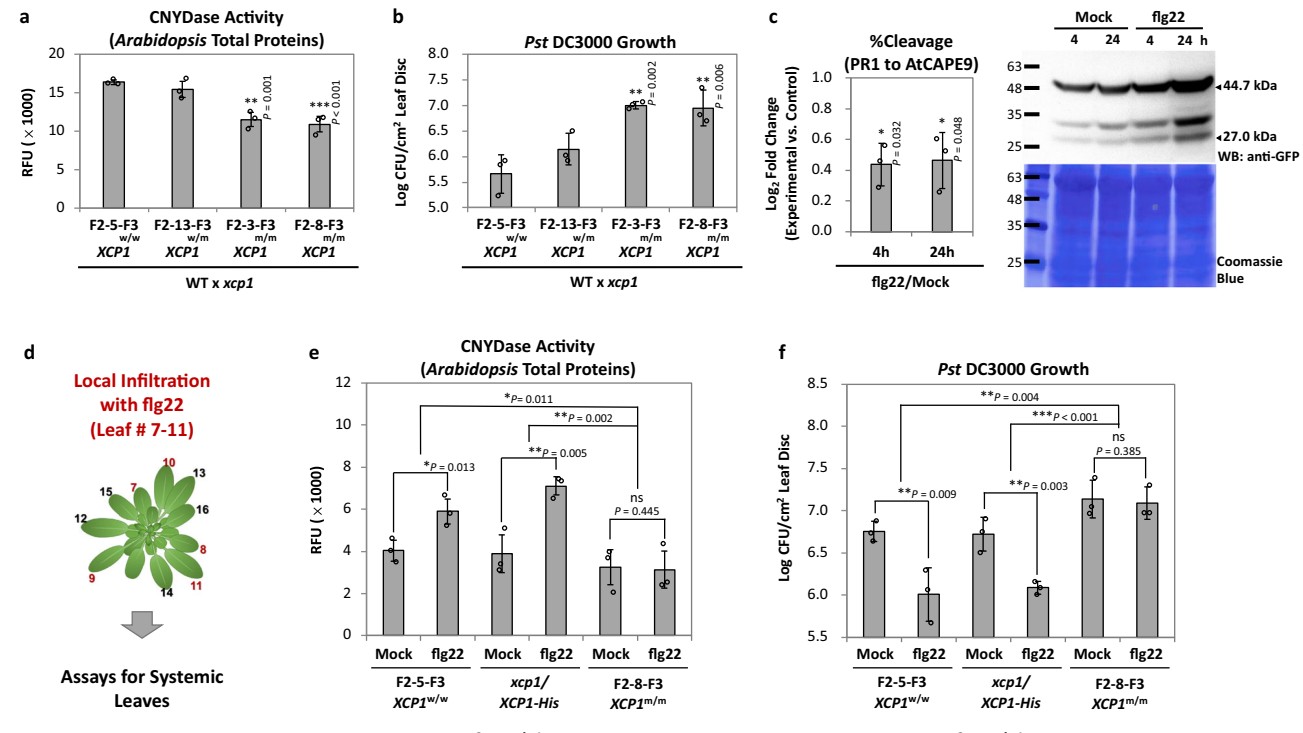

**Fig. 6 | Role of XCP1 in regulating *Arabidopsis* immunity. a** CNYDase activity of *Arabidopsis* protein extract and **b** *Pst* DC3000 growth using the F3 generation of WT and *xcp1 Arabidopsis* crossed plant (WT × *xcp1*) lines, including homozygous wild-type *XCP1/XCP1* (*XCP1*^w/w, F2–5–F3), heterozygous mutated *XCP1* (*XCP1*^w/m, F2–13–F3) and homozygous mutated *XCP1* (*XCP1*^m/m, F2–3–F3 and F2–8–F3) lines. Log colony-forming units (Log CFU) of *Pst* DC3000 were measured after 7 days of inoculation. Values are means ± SD of three biological replicates. Each replicate was obtained from the pooling of three plants. **c** Left panel: The cleavage percentage of PR1-eYFP for AtCAPE9-eYFP production (%Cleavage) in transgenic *PR1-eYFP Arabidopsis* seedlings treated by flg22 for 4 or 24 h. The %Cleavage was calculated from the band intensity of AtCAPE9-eYFP divided by the sum of PR1-eYFP and AtCAPE9-eYFP band intensities detected by immunoblot using anti-GFP (Supplementary Fig. 14). Values are means ± SD of three biological replicates. Each replicate was obtained from pooling of six plants. Right panel: Immunoblot of PR1-eYFP and AtCAPE9-eYFP fragment in transgenic PR1-eYFP *Arabidopsis* treated by Mock or

flg22 for 4 or 24 h. The sizes of intact PR1-eYFP and AtCAPE9-eYFP were estimated to be ~44.7 kDa and ~27.0 kDa, respectively, detected by the anti-GFP. Coomassie Blue staining shows total protein loaded. **d** Assays for the systemic leaves of *Arabidopsis* plants with local pre-infiltration of flg22. The local pre-infiltration was performed on leaves # 7–11, and the other leaves without infiltration were collected as systemic leaves of each plant. **e** The CNYDase activity and **f** the growth of *Pst* DC3000 in the systemic leaves of *XCP1*^w/w (F2–5–F3), the *xcp1* complemented by expression *XCP1-His* (*xcp1/XCP1-His*), and *XCP1*^m/m (F2–8–F3) lines of *Arabidopsis* locally treated with flg22. Log colony-forming units (Log CFU) of *Pst* DC3000 were measured after 5 days of inoculation. Each proteolytic activity assay was performed by incubating 50 µg protein extract with 25 µM substrate. Relative fluorescent units (RFU) of the cleaved fluorophore were measured after 5 h incubation with the substrate. Values are means ± SD of three biological replicates. Each replicate was obtained by pooling plant tissues of three plants. *P* values were calculated by one-tailed unpaired *t*-test (ns, no significant change; *, *P* < 0.05; **, *P* < 0.01; ***, *P* < 0.001).

followed by incubation Ni-NTA-purified proteins from *N. benthamiana* without or with expressing XCP1-His or XCP1^C161A-His. Incubation with purified XCP1-His increased the release of AtCAPE9-eYFP from immobilized PR1-eYFP compared to the controls (Ni-NTA-purified protein from *N. benthamiana* expressing P19 along or P19 plus XCP1^C161A-His) (Fig. 5g). As PR1 has been shown as a secretory protein localized in apoplast[22–25] and XCP1 has been detected in xylem extract[19,25–27], we further examined whether the XCP1 can also localized in apoplast like PR1. We respectively expressed the XCP1-GFP and PR1-eYFP proteins in *N. benthamiana* leaves using *Agrobacterium* transformation and found that both XCP1-GFP and PR1-eYFP were localized in the apoplast by referring to the plasma membrane location using PIP2A-mCherry[28] (Supplementary Fig. 12). These data suggest that XCP1 directly processes PR1 into AtCAPE9.

## XCP1 regulates local and systemic immunity in *Arabidopsis*

Our data so far suggest that XCP1 is ESCAPE that can generate the phytocytokine AtCAPE9 from PR1, so we further investigated how this enzyme regulates CNYDase activity and disease resistance of *Arabidopsis*. Briefly, we genotyped F2 generation of WT and *xcp1* crossed plant (WT × *xcp1*) lines and cultivated specific F2 lines F2–5, F2–3, F2–8, and F2–13 for F3 homozygous WT *XCP1* (F2-5-F3 *XCP1*^w/w) and mutated

*XCP1* (F2–3–F3 *XCP1*^m/m and F2–8–F3 *XCP1*^m/m), and heterozygous mutated *XCP1* (F2–13–F3 *XCP1*^w/m) plants for CNYDase and pathogen infection assays (Supplementary Fig. 13). We found that the reduction of the CNYDase activity and pathogen resistance in *XCP1*^m/m lines as compared to *XCP1*^w/w and *XCP1*^w/m lines (Fig. 6a, b). To examine pathogen-mediated regulation of PR1 proteolytic processing, we applied a conserved pathogen-associated molecular pattern (PAMP) elicitor, flg22, to plants expressing *PR1-eYFP*. Flg22 treatment for 4 and 24 h significantly promoted the proteolysis of PR1-eYFP (~44.7 kDa) and, in turn, the production of AtCAPE9-eYFP (~27.0 kDa) (Fig. 6c; Supplementary Fig. 14), suggesting that flg22 treatment promotes the cleavage of PR1 to produce AtCAPE9. To show that XCP1 is indeed involved in systemic immunity triggered by PAMP, we treated plants locally with flg22 and then monitored CNYDase activity and disease resistance in untreated systemic leaves. Specifically, we employed F2–5–F3 *XCP1*^w/w and F2–8–F3 *XCP1*^m/m, and a stable complement line was obtained by expressing *XCP1-His* in the *xcp1* mutant (*xcp1/XCP1-His*). We locally infiltrated flg22 and observed reduced CNYDase activity and diminished resistance to *Pst* DC3000 in the untreated leaves of the *XCP1*^m/m line compared to both *XCP1*^w/w and *xcp1/XCP1-His* lines (Fig. 6d–f). As flg22 eliciting SAR depends on the biosynthesis and perception of SA, we further tested the role of AtCAPE9 in the SA-

mediated systemic resistance. Here, we compared how local flg22 or AtCAPE9 triggers plant SAR against *Pst* DC3000 in WT, SA-biosynthesis mutant *ics1* and SA-perception mutant *npr1* plants (Supplementary Fig. 15). As expected, flg22-elicited plant SAR was significantly abolished in *ics1* and *npr1*. However, AtCAPE9 can still trigger SAR without the presence of *ICS1* and *NPR1*.

## Discussion

In this study, we identified endogenous AtCAPE9 and showed its bioactivity in activating SA biosynthesis and systemic immune response in *Arabidopsis*. We show for the first time that AtCAPE9 is derived from the cleavage of the putative substrate domain CNYD in PR1 in SA-treated *Arabidopsis* leaves. The aspartate residue (D150) of the CNYD domain is crucial for the cleavage of PR1-eYFP to generate AtCAPE9-eYFP in plants, indicating that AtCAPE9 production requires an aspartate-specific protease. We demonstrated that both the CNYD and CAPE domains in PR1 are crucial for systemic immunity by measuring the SAR response in *pr1* with locally-expressed native or CNYD/CAPE-mutated PR1 using *Agrobacterium*. Under local infection of *Agrobacterium*, the induced SAR in the WT plant can also be observed in the *pr1* expressing native *PR1* but not in the *pr1* expressing CNYD/CAPE-mutated *PR1*. Further, the treatment of AtCAPE9 to the local leaves of *pr1* directly enables systemic immunity without *Agrobacterium* infection, which implies AtCAPE9 may directly or indirectly elicit systemic immunity. Since SA is heavily involved in plant systemic immunity, we also showed that SA treatment can enhance the endogenous production of AtCAPE9, the proteolytic cleavage of PR1-eYFP to produce AtCAPE9-eYFP, and CNYDase activity in plants. Together with our observation, SA both induces and is induced by AtCAPE9, suggesting the existence of a positive-feedback loop to amplify SA-mediated systemic defense responses in plants.

We successfully identified an *Arabidopsis* PLCP member XCP1 as an enzyme specific for CAPE production (ESCAPE). XCP1 has been reported to aid in the formation of tracheary elements (TEs) prior to the macro-autolysis, followed by vacuole and protoplast disruption[26]. XCP1 is localized in the vacuole and released to the xylem during the formation of TEs. The XCPs loss-of-function plants *xcp1xcp2* show a delay of micro-autolysis but do not exhibit a noticeable change in normal growth and developmental phenotypes, including TEs, as compared to WT plants. Although XCPs are not essential for *Arabidopsis* growth and development, XCP1 has been shown to participate in basal defense functions in alternative systems[29,30], suggesting the XCP1-CAPE signaling may be involved in other plants or the infection by other pathogens. When expressing *Cladosporium fulvum* effector Avr2 in *Arabidopsis*, it was found to bind the XCP1 protein and inhibit its cysteine protease activity[29]. Therefore, Avr2-expressing plants exhibiting higher susceptibility to fungal infection may be caused by Avr2 targeting XCP1 to disrupt CNYDase activity and, in turn, the production of AtCAPE9 and the induction of SA. A more recent study indicates that the pathogen effector Fol-Secreted virulence-related protein1 (FolSvp1) from the fungal pathogen *Fusarium oxysporum* targets tomato PR1 (SlPR1) for translocating secreted SlPR1 back into the tomato cells[30]. This mechanism caused by fungal effector largely suppresses the cleavage of PR1 to produce phytocytokine CAPE, and the suppression of CAPE from PR1 effectively reduces the tomato resistance to *Fusarium oxysporum*. Taken together, the identification of XCP1 as ESCAPE highlights the importance of CAPE production in plant resistance against pathogens and suggests that plant pathogens can use diverse strategies to suppress CAPE production for their virulence. Given that both PR1 and XCP1 are conserved in plant kingdom[31,32], this mechanism may be widespread among plant systems.

XCP1 plays a major role in the CNYDase activity in plants, given that protein extract from *xcp1* mutants was not tagged with biotin−CNYD−CHO and showed a significant loss of CNYDase activity.

However, it is still possible that the residual CNYDase activity and SAR response observed in *xcp1* may be contributed by XCP2 or other unidentified proteases, as neither *xcp1* nor *xcp2* single mutant shows completely abolished lysate CNYDase activity compared to the WT. Further, the mutation of *XCP1* indeed reduced the in vivo production of endogenous AtCAPE9-eYFP in the plant overexpressing PR1-eYFP and AtCAPE9 from native PR1. Purified XCP1-His was highly specific to the CNYD domain, and its CNYDase activity was enhanced by $Ca^{2+}$ but repressed by $Zn^{2+}$ and E-64. Importantly, we provide strong evidence that purified XCP1-His directly cleaves PR1-eYFP to release AtCAPE9-eYFP. XCP1 activity was temperature sensitive, displaying optimal CNYDase activity at 22 °C, which is also optimal for the growth of *Arabidopsis*. In addition, XCP1 displayed a $K_m$ of 26.5 μM for the CNYD substrate, which is comparable to the $K_m$ for human caspase-1 with the YVAD substrate (20 μM)[33]. Interestingly, the optimal pH for XCP1 CNYDase activity was altered by the addition of excess $Ca^{2+}$, indicating that pH and $Ca^{2+}$ levels can fine-tune the production of AtCAPE9 by XCP1 to regulate plant immunity. In the cytosol, the pH is buffered at 7 with low $[Ca^{2+}]$ levels ranging from 1 to 200 nM[34]. Under these conditions, XCP1 exhibits its lowest CNYDase activity, which could prohibit the production of CAPE in the cytosol. However, the presence of excess $Ca^{2+}$ leads to highly activated CNYDase activity in XCP1 at pH 6.0. Given that both PR1 and XCP1 are mainly expressed in the apoplast/xylem, where $[Ca^{2+}]$ levels are normally >1 mM and pH ranges from 5 to 7[35], the production of AtCAPE9 by XCP1 likely occurs in the apoplast. This reaction could be manipulated not only by the expression of PR1 and XCP1 but also by the cellular transporters that shape the apoplastic $pH/[Ca^{2+}]$. Thus, the regulation of pH and $[Ca^{2+}]$ in the apoplast is likely to play a crucial role in modulating the production of AtCAPE9 and in the resistance of plants against pathogen attack. A study suggested that the processing of PR1 to produce CAPE in tomatoes occurs in the apoplast[30] could also support the proposed mechanism. We showed that the CNYDase activity of XCP1 is dependent on C161 in its putative protease active site. The higher pathogen susceptibility of the *xcp1* than the lines with *XCP1* expressed indicates the function of *XCP1* in resisting pathogen infection. Similar to the treatment of SA and INA, the PAMP flg22 enhanced the cleavage of PR1-eYFP to produce AtCAPE9-eYFP. We determined that the flg22-elicited systemic immunity is mediated by XCP1, according to the observation that systemic CNYDase activity and antipathogen responses were diminished in *xcp1* but can be recovered by complementing the *XCP1-His*. Flg22 is unable to elicit SAR in the absence of *XCP1*, suggesting that the positive signaling feedback loop that amplifies SA production for SAR may be blocked in the *xcp1*, likely due to the reduced production of AtCAPE9. We further demonstrated that the induction of SAR by AtCAPE9 can be independent of SA-signaling, indicating that AtCAPE9 can be downstream of SA-signaling in this mechanism. These discoveries also enable us to propose a model for the systemic immunity elicited by the PAMP flg22 (Fig. 7). As flg22 activates the SA biosynthesis and SA perception by NPR1 for PR1 expression has been well determined[36,37], our study discovered flg22/SA further enable the activation of CNYDase activity of XCP1 for the production of an essential SAR signal, AtCAEP9. Flg22/SA may regulate the protein expression and/or tune the $pH/Ca^{2+}$ for activating the CNYDase activity of XCP1, which can be compromised by high temperature. It is highly possible that AtCAPE9 not only forms a positive signaling feedback loop with SA but can also function as a mobile signal or elicit other signals for SAR. As pipecolic acid (Pip)/N-hydroxy-pipecolic acid (NHP) has been suggested to form a positive feedback loop with SA[38,39] and elicit the pathway alternative to SA[40,41], AtCAPE9 may induce or be induced by Pip/NHP for SAR.

In summary, XCP1 is a plant protease for initiating systemic immunity. The enzyme activity of XCP1 is cysteine-dependent, is specific for cleaving the aspartate of a four amino acid substrate domain, and can release the cytokine AtCAPE9 to activate systemic immunity.

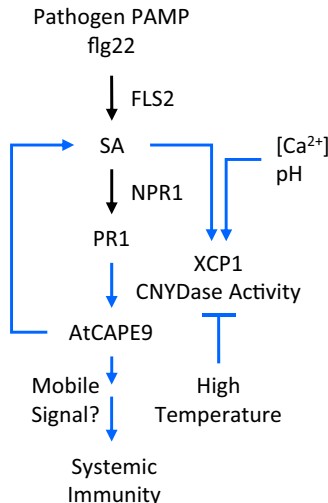

**Fig. 7 | Proposed model of XCP1-processing PR1 to AtCAPE9 for the activation of plant systemic immunity triggered by pathogen PAMP flg22.** The blue lines indicate the regulation mechanisms uncovered in this study.

The study suggests that AtCAPE9 is the primary CAPE peptide produced by XCP1 for SAR, as it is the only CAPE member detected in SA-treated leaves, and mutating the CNYD/CAPE domain of PR1 resulted in the loss of SAR response. As the secretion of PR1 is thought to be crucial for SAR[25] and we detected XCP1 can be localized in the apoplast, suggesting that PR1 is processed by XCP1 in the apoplast to induce SAR. Previous studies identified XCP1 in the xylem sap of *Brassica oleracea*[42], together with our data, suggesting that XCP1 serves a systemic immune regulatory role for PR1 processing through the xylem sap for long-distance migration. The temperature sensitivity of XCP1 activity and SA induction of AtCAPE9 might help to explain why pathogen-induced SA-defense responses are temperature-vulnerable[43]; lower XCP1 activity at higher temperatures can yield less AtCAPE9 and thus diminished SA biosynthesis. Analog to animal caspases that activate and secret immunomodulatory cytokines, the plant can utilize a caspase-like protease XCP1 to target and proteolyze the PR1 as pro-cytokine for releasing the phytocytokine AtCAPE9 for systemic immunity. Our findings suggest that the sensitization of the CAPE production mechanism during pathogen infection may help to engineer crops more resistant to diseases.

## Methods

### Plant materials and growth conditions

The WT plant seeds of *Arabidopsis thaliana* Columbia (Col-0, CS70000) and all T-DNA insertion mutant lines were purchased from the Arabidopsis Biological Resource Center (ABRC). All plant seeds used in this work were subjected to surface sterilization before use. The *Arabidopsis* plants grown in soil were cultivated in a growth chamber with a 10/14 h light/dark cycle (for vegetative growth and delayed flowering) or a 16/8 h light/dark cycle (for seed collection) at 22 °C under light and 20 °C under dark. The *Arabidopsis* seedlings were grown by sowing the seeds on half-strength Murashige & Skoog (½ MS) agar plates containing 0.44% (w/v) MS salts, 1% (w/v) sucrose, 0.05% (w/v) MES and adjusted with KOH to pH 5.7. These seeds on plates were incubated at 4 °C for 2 days in the dark, then cultivated at 22 °C in a growth chamber under a 16/8 h light/dark cycle for 10 days before use. Eight-week-old Col-0 plants grown in soil were used for quantifying the SA level regulated by AtCAPE9. Twelve-day-old Col-0 seedlings grown in ½ MS agar plates were used for examining the plant pathogen resistance enhanced by AtCAPE9. Nine-week-old Col-0 and *xcp1* mutant plants grown in soil were used for the detection and quantification of endogenous AtCAPE9. Twelve-day-old transgenic

*Arabidopsis* seedlings grown in ½ MS agar plates expressing native (N) and alanine-substituted (D150A and P151A) PR1 fused to a C-terminal enhanced yellow fluorescent protein (eYFP) were used for monitoring the cleavage of PR1. *Nicotiana benthamiana* (*N. benthamiana*) grown in soil at 25 °C in a walk-in growth chamber under a 16/8 h light/dark cycle for 4 weeks was used for transient expression.

The F2 homozygous WT *XCP1* (*XCP1^{w/w}*, F2–5), the F2 heterozygous mutated *XCP1* (*XCP1^{w/m}*, F2–13) and two F2 homozygous mutated *XCP1* lines (*XCP1^{m/m}*, F2–3 and F2–8) were generated by crossing WT and T3 generation of *xcp1* lines (WT × *xcp1*). Genotyping results of the *XCP1* zygosity in WT × *xcp1* F2 lines and F3 plants obtained from self-crossing of F2–3, F2–5, F2–8, and F2–13 lines were illustrated in Supplementary Fig. 13a. T3 generation of *XCP1-His* complemented *xcp1* plant (*xcp1/XCP1-His*) were generated by expressing native promoter-driven *XCP1-His* in *xcp1*. Genotyping and expression of *XCP1* in *xcp1/XCP1-His* were illustrated in Supplementary Fig. 13b, c. 6-week-old F3 generation of WT × *xcp1* plants and T3 generation of *xcp1/XCP1-His* grown in soil was used to examine the CNYD-targeted protease (CNYDase) activity and plant resistance to the pathogen. *xcp1* × *PR1-eYFP* cross lines were generated by crossing of *xcp1* and *PR1-eYFP*-overexpressing WT. Expression of *XCP1* and *PR1-eYFP* in *xcp1* × *PR1-eYFP* F2 lines was confirmed in Supplementary Fig. 7a. Twelve-day-old *xcp1* × *PR1-eYFP* F3 plants were used for monitoring the cleavage of PR1. The *pr1* mutant was generated applying CRISPR/Cas9, genotyped, and sequenced to confirm the mutated sequence of *PR1* (Supplementary Fig. 2a, b). The RNA expression and protein expression were confirmed by RT-PCR and western blotting, respectively (Supplementary Fig. 2c, d). Six-week-old *pr1* T3 plants were used to transiently express the different forms of *PR1* and examine plant resistance to pathogens. The RNA expression and protein expression in the *pr1* expressing various sequences of *PR1* transiently were monitored by RT-PCR and western blotting, respectively (Supplementary Fig. 2d, e).

All T-DNA insertion mutants of *Arabidopsis* used in this study are shown in Supplementary Table 2. All mutants, transgenic lines, and cross lines were genotyped for zygosity using allele-specific primers shown in Supplementary Table 3 and Phire Plant Direct PCR Master Mix (#F160L, Thermo Fisher Scientific).

### RNA isolation, RT-PCR, and qRT-PCR

Total RNA was isolated from 12-day-old seedlings for RT-PCR and qRT-PCR analysis, and 6-week-old plant leaves for RT-PCR analysis using a Total RNA mini kit (#RP300, Geneaid). cDNA was synthesized from 5 µg total RNA using SuperScript III reverse transcriptase (#18080085, Invitrogen). The RT-PCR was performed using 2× SuperGreen PCR Master Mix (#TTC-PA31-5, Biotools) with 1 µL of each cDNA sample for 25 cycles. The qRT-PCR was performed using Fast SYBR™ Green Master Mix (#4385612, Thermo Fisher Scientific) and ABI 7500 Fast Real-Time PCR Systems. The PCR program was initiated with the holding stage at 95 °C for 20 s, followed by the cycling stage at 95 °C for 3 s and at 60 °C for 30 s, with a total of 40 cycles. The gene expression of different samples was normalized with internal control *ACTIN2*. The primers used are listed in Supplementary Table 3.

### DNA construction and cloning

All primers used for cloning are listed in Supplementary Table 3. To generate the *pr1* mutant, three gRNA sequences: gRNA 1 (5′-TAGCC-CACAAGATTATCTAA-3′), gRNA 2 (5′-TCCCATGCAGTGGGACGAGA-3′) and gRNA 3 (5′-GCAGACTCATACACTCTGGT-3′) using CRISPR-P 2.0 (http://cbi.hzau.edu.cn/CRISPR2/) and RNAfold web server (http://rna.tbi.univie.ac.at/cgi-bin/RNAWebSuite/RNAfold.cgi) were designed and selected and then cloned into pHEE401E vector[44].

To obtain the target sequences of *PR1*, AtCAPE9-truncated *PR1* (*PR1^{ΔCAPE9}*), and *XCP1*, the cDNA sequences of *Arabidopsis* were produced via reverse transcription, and the DNA fragments of *PR1* and *PR1^{ΔCAPE9}* were amplified by the specific primer using 2× Super Hi-Fi Taq

PCR MasterMix (KTT-BB05, Biotools). The *XCP1* DNA fragment was amplified by Perfectread *Pfu* polymerase (PE-P101, Ten Giga Bio). To generate the clone of alanine-substituted D150A-PR1 (*PR1^D150A*), the construct of pUC57-*p35S::PR1^D150A* was synthesized and purchased from Genomics Co., Ltd. The DNA fragment of *PR1^D150A* was amplified by the specific primer using 2× Super Hi-Fi Taq PCR MasterMix. The final product of *XCP1* has added a DNA sequence for the C-terminal 6× Histidine tag (HisTag) using a specific primer for purification purposes.

The DNA products of *PR1*, *PR1^D150A*, *PR1^ΔCAPE9*, *XCP1* and *XCP1-His* were cloned into the pCR8 entry vector using pCR™8/GW/TOPO™ TA Cloning Kit (#450642, Invitrogen). The pCR8 clones with target genes were transferred into a gateway vector pMDC32 carrying a dual 35S promoter. The clones of pMDC32-*2x35S::PR1*, pMDC32-*2x35S::PR1^D150A*, pMDC32-*2x35S::PR1^ΔCAPE9*, pMDC32-*2x35S::XCP1-His* and modified pMDC32-*pXCP1::XCP1-His* were generated. The C161A mutated *XCP1-His* (*XCP1^C161A-His*) clone used for in vitro functional study was obtained by pMDC32-*2x35S::XCP1-His* using a pair of point mutation primers and then cloned into pMDC32 to obtain the pMDC32-*2x35S:: XCP1^C161A-His*. The pCR8-*XCP1* was cloned to the pMDC83 carrying the C-terminal GFP for checking the subcellular localization. The native and two alanine-substituted (N, D150A and P151A) *PR1* clones were transferred into pK7YWG2 carrying a single 35S promoter, using a method outlined in our previous study[45].

### Agrobacterium-mediated transformation

For expressing the chosen genes in plants, the *Agrobacterium*-mediated transformation was performed. All the binary vectors were transformed into *Agrobacterium tumefaciens* strain GV3101 using electroporation.

*Arabidopsis thaliana* transgenic plants were generated by the floral dipping method[46]. To observe the proteolytic processing of PR1, the N, D150A, and P151A PR1-eYFP clones were separately transformed into *Arabidopsis* WT plants to produce a stable transgenic line (T3 generation homozygous lines were used in this study). To generate a *pr1* mutant using CRISPR/Cas9 system, the gRNA-containing pHEE401E was transformed using *Agrobacterium* into WT plants to produce a stable *pr1* mutant line with the truncated editing of *PR1* (T3 generation homozygous lines were used in this study). To generate the *XCP1* complementation lines, we expressed the *XCP1-His* gene driven by the native promoter in the T3 generation of the *xcp1* mutant (T3 generation homozygous lines were used in this study). The expression of *XCP1* in the WT, *xcp1*, and *xcp1/XCP1-His* plants was quantified by RT-PCR (Supplementary Fig. 13c).

For the transient gene expression in *Arabidopsis* leaves, *Agrobacteria* carrying plasmids with chosen genes and the *Agrobacterium* carrying pBIN61-*P19* were cultured in liquid LB broth containing antibiotics at 28 °C for 3 days. The bacterial broth was transferred to LB medium containing 10 mM MES, pH 5.7, 20 μM acetosyringone (AS), and antibiotics (kanamycin 50 mg/L, spectinomycin 100 mg/L, rifampicin 50 mg/L), then cultured at 28 °C for overnight. After, the bacteria were pelleted by centrifugation at 5000 × *g* for 5 min and resuspended by infiltration buffer (10 mM MgCl₂, 10 mM MES, pH 5.7, 1% sucrose, 0.01% Silwet L-77 and 200 μM AS) with OD₆₀₀ = 1.0 and incubated for 2 h. *Agrobacteria* carrying different plasmids were mixed with the *P19*-carrying *Agrobacterium* in a 1:1 ratio (v/v) and infiltrated into leaves of 6-week-old *Arabidopsis* leaves with 1 mL plastic syringes. The infiltrated plants were kept under the light for 1 h to dry the leaves, under the dark for 24 h, and then kept with a 16/8 h light/dark cycle at 22 °C under light and 20 °C under dark for 4 days. To transiently express the different forms of *PR1* in the *pr1* mutant, the plasmids pMDC32-*2x35S::PR1*, pMDC32-*2x35S::PR1^D150A*, and pMDC32-*2x35S::PR1^ΔCAPE9* were transiently expressed in the locally selected leaves of *pr1* mutant plants.

For the transient gene expression in *N. benthamiana* leaves, the procedures were slightly modified from the steps for the expression in *Arabidopsis*. Briefly described, after the culture, the bacteria were pelleted and resuspended by infiltration buffer (10 mM MgCl₂, 10 mM MES, pH 5.7, and 200 μM AS) with OD₆₀₀ = 0.5 and incubated for 2 h. After the infiltration, plants were kept at 25 °C with a 16/8 h light/dark cycle for 2-3 days. To express the XCP1-His and XCP1^C161A-His protein, the *2x35S::XCP1-His* or *2x35S::XCP1^C161A-His* together with *P19* were transiently expressed. To examine the subcellular localization of XCP1 and PR1, *2x35S::XCP1-GFP* or *35S::PR1-eYFP* were transiently expressed along with *35S::PIP2A-mCherry*.

### Purification of His-tagged protein by Ni–NTA affinity chromatography

Proteins were extracted from *XCP1-His* and *XCP1^C161A-His*-over-expressed *N. benthamiana* leaves and frozen in liquid nitrogen and ground into a fine powder, then dissolved in the plant extraction buffer (50 mM sodium acetate with 200 mM NaCl and 3 mM DTT at pH 5.0) and rotated at 50 rpm for 30 min at 4 °C. The protein extract was then centrifuged at 16,000 × *g* at 4 °C for 15 min, and the supernatant was filtered by 70 μm Nylon cell strainers (Falcon). The filtered supernatant was precipitated by acetone at −20 °C overnight to remove DTT. The mixture was centrifuged at 16,000 × *g* at 4 °C for 15 min to remove acetone, and the protein pellet was air-dried in the laminar flow for 30 min. The protein pellet was resuspended by Ni-NTA binding buffer (50 mM monobasic sodium phosphate, 500 mM NaCl, 10 mM imidazole, pH 8.0), rotated at 50 rpm at 4 °C for 30 min, and then centrifuged at 16,000 × *g* at 4 °C for 5 min. The supernatant was collected and loaded into a Ni-NTA agarose (#30230, QIAGEN) packed column, which was pre-equilibrated with Ni–NTA binding buffer. His-tagged protein was washed in a buffer containing 20 mM imidazole (pH 8.0) and eluted in a buffer containing 250 mM imidazole (pH 8.0). Each step during the purification was analyzed by western blotting with anti-His antibody (mouse, 1:5000; #PPT-66005-1, Biotools) (Supplementary Fig. 11b).

### Protein subcellular localization

The subcellular localization of XCP1 and PR1 was examined by transforming the *2x35S::XCP1-GFP* or *35S::PR1-eYFP* along with *35S::PIP2A-mCherry* in *N. benthamiana* leaves using *Agrobacterium* infiltration. After 48 h of infiltration, the leaves were treated with 1 M mannitol for the plasmolysis and then observed by the Zeiss Axio Imager Z1 microscopy (Carl Zeiss). The fluorescence of GFP and eYFP was detected at 460–500 nm excitation and 510–560 nm emission, whereas the fluorescence of mCherry was detected at 510-560 nm excitation and 574-648 nm emission.

### Peptide or phytohormone treatment

For detecting the SA concentrations regulated by AtCAPE9, 8-week-old Col-0 plants were sprayed with water or an aqueous solution of 250 nM synthetic AtCAPE9 (purity >95%, Mission Biotech). For measuring the bacterial growth of the plants treated with AtCAPE9, 12 days-old Col-0 seedlings were immersed in ½ MS medium added with water or an aqueous solution of 250 nM synthetic AtCAPE9 for 6 h in the dark before pathogen inoculation. To detect and quantify endogenous AtCAPE9 production, 9-week-old Col-0 plants untreated or treated with an aqueous solution of 1 mM SA containing 0.0015% Silwet L-77 were used. To monitor the proteolytic processing of PR1 regulated by 2,6-dichloroisonicotinic acid (INA) and SA, 12-day-old seedlings of *PR1-eYFP* transgenic lines were immersed in 10 mM MgSO₄ (Mock), 60 μM SA in 10 mM MgSO₄, 60 μM INA in 10 mM MgSO₄ for 5, 30 and 60 min. To monitor the proteolytic processing of PR1 regulated by flg22, 12-day-old seedlings of *PR1-eYFP* transgenic lines were immersed in 10 mM MgSO₄ (Mock) or 500 nM synthetic flg22 (purity >95%, Mission Biotech) in 10 mM MgSO₄ for 4 and 24 h. To monitor the proteolytic processing of PR1 regulated by INA under different temperatures, 12-day-old seedlings of *PR1-eYFP* transgenic lines were pre-

incubated at 32 or 22 °C for 30 min and then immersed in 10 mM MgSO₄ (Mock) or 60 µM INA in 10 mM MgSO₄ for 60 min. To monitor the CNYDase activity triggered by wounding, INA, or SA in plants, 8-week-old Col-0 plants were wounded by forceps, sprayed with 10 mM MgSO₄ (Mock), 60 µM SA in 10 mM MgSO₄ or 60 µM INA in 10 mM MgSO₄ for 24 h. To examine the role of XCP1 in flg22-triggered systemic immunity, the selected leaves of plants were infiltrated with 10 mM MgSO₄ (Mock) or 500 nM flg22 in 10 mM MgSO₄ for 48 h, and their corresponding untreated leaves were used as the systemic leaves.

## Pathogen infection

The bacterial pathogen *Pseudomonas syringae* pv. *tomato* DC3000 (*Pst* DC3000) was grown on King's B (KB) agar medium containing 50 mg/L rifampicin at 28 °C for 2 days and then cultured in KB liquid medium at 28 °C with 230 rpm shaking overnight. The bacteria were pelleted by centrifugation and resuspended in 10 mM MgSO₄ containing 0.005% Silwet L-77 with OD₆₀₀ = 0.02. 12-day-old seedlings treated with the mock or AtCAPE9 peptide solution for 6 h were then dipped in the bacterial suspension for 7 days. 6-week-old WT and *pr1* locally infiltrated with different agents for 4 days, 6-week-old F3 WT × *xcp1* or *xcp1/XCP1-His* plants locally infiltrated with the mock or 500 nM flg22 for 48 h, 6-week-old WT, *ics1*, and *npr1* locally infiltrated with the 500 nM flg22 or AtCAPE9 for 48 h, and 9-week-old F3 WT × *xcp1* lines without treatments were dipped into the bacterial suspension for 1 min and then kept for 5 or 7 days with a 10/14 h light/dark cycle. The bacterial populations were measured from the extracts of seedlings or adult plants after 2 days of inoculation on the KB agar medium, represented as log colony-forming units (Log CFU) per seedling or per leaf disc (cm²) according to a previous method[13].

## Endogenous peptide isolation and detection

Nine-week-old Col-0 plants untreated or treated with an aqueous solution of 1 mM SA containing 0.0015% Silwet L-77 were collected and individually ground into powder under liquid nitrogen using a homogenizer (Nissei ACE Homogenizer AM-5). The frozen leaf powder was dissolved in 1% TFA (using 3 mL of solvent for each gram of leaves powder) and homogenized to leaf juice using a blender for 2 min. The leaf juice was filtered through four layers of cheesecloth and one layer of Miracloth (#475855-1 R, Calbiochem). The filtrated leaf juice was then centrifuged at 10,000 × *g* for 20 min at 4 °C. The supernatant was adjusted to pH 4.5 with 10 N NaOH and centrifuged at 10,000 × *g* for 20 min at 4 °C. Then the supernatant was re-adjusted to pH 2.5 with 6 N HCl. The tryptic β-casein peptides were purified using a C18 Oasis HLB 1cc (10 mg) cartridge (#186000383, Waters) and added into peptide extract as an internal control. To purify the supernatant from the extract of 50 g leaf tissue, the customized C18 Sep-pak (20 g) cartridge (#WAT020585, Waters) was used and first equilibrated by 60 mL 0.1% TFA. The supernatant from 50 g tissue was loaded into the Sep-Pak cartridge, washed with 100 mL of 0.1% TFA and eluted with 150 mL of 60% methanol in 0.1% TFA. To purify the supernatant from the extract of 5 g leaf tissue, the C18 Sep-pak (5 g) cartridge (#WAT036925, Waters) was used and first equilibrated by 20 mL of 0.1% TFA. The supernatant from 5 g tissue was loaded into the Sep-Pak cartridge, washed with 40 mL of 0.1% TFA and eluted with 35 mL of 60% methanol in 0.1% TFA. The eluted solution was vacuum-evaporated using a vacuum centrifugation concentrator (miVac Duo Concentrator, Genevac) to dryness. The dried crude extract was dissolved in 1 mL of 0.1% TFA, centrifuged at 10,000 × *g* for 10 min at 4 °C and filtered through a 0.2 µm filter (#2401, PALL, 0.13 mm wwPTFE) before peptide fractionation. The filtrated samples were injected into a Superdex Peptide 10/300 column (#GE17-5176-01, GE Healthcare) operated by an FPLC AKTA Pure 25 (GE Healthcare) and eluted by 0.5 mL/min of 0.1% TFA with 1 fraction/min for collecting the peptide fractions. The collected fractions were evaporated to dryness by a vacuum centrifugal concentrator and then desalted by C₁₈ ZipTip (#ZTC18S960, Merck

Millipore) before LC−MS/MS analysis. To identify endogenous AtCAPE9, the LC−MS/MS operated in data-dependent acquisition (DDA) mode was used. To quantify the abundance of AtCAPE9 in *Arabidopsis*, the LC−MS/MS operated in parallel reaction monitoring (PRM) MS/MS mode targeting precursor and specific fragment ions of AtCAPE9.

## Phytohormone extraction

The metabolites extraction procedure was modified from a previously published protocol[47]. The leaf tissues (about 0.4 g fresh weight) were ground into a powder under liquid nitrogen and transferred to a 50 mL screw-cap tube (#3181-345-008-9, Labcon). The frozen leaf powder was dissolved in 4 mL extraction solvent, and d₆-SA (#616796, Sigma-Aldrich) (2 ng for 0.4 g leaf tissue) was added as internal standards. The samples were extracted by rotating at a speed of 100 rpm at 4 °C for 30 min, and then 8 mL dichloromethane was added to each sample and shaken at 100 rpm at 4 °C for 30 min. The samples were centrifuged at 13,000 × *g* at 4 °C for 15 min, and two phases were formed. The lower phase was transferred carefully into a new tube and evaporated to dryness by a vacuum centrifugal concentrator. The dried samples were dissolved in 300 µL methanol, mixed well, and centrifuged at 10,000 × *g* at 4 °C for 5 min, and then the supernatant was transferred to the sample vial (#186002639, Waters) for targeted quantitation analysis using LC−MS/MS.

## Targeted peptide and phytohormone quantitation using LC-MS/MS

To identify and quantify AtCAPE9 in untreated and SA-treated WT samples, a linear ion trap-orbitrap mass spectrometer (Orbitrap Elite, Thermo Fisher Scientific) coupled online with a nanoUHPLC system (nanoACQUITY UPLC, Waters) was used. For peptide abundance normalization, a 50 µg tryptic β-casein peptide mixture as an internal standard was added to the sample before SPE purification. Samples were loaded into an analytical column (BEH130 C18, 1.7 µm, 75 µm × 250 mm, Waters) and were separated using a 68 min linear gradient from 8 to 30% ACN with 0.1% formic acid at 300 nL/min flow and a column temperature of 35 °C. The mass spectrometer was operated in positive ion mode and set to one full FT-MS scan (m/z 400−1600) with 60,000 resolution, followed by ion trap MS/MS full scans for AtCAPE9 and internal standard quantification. For MS/MS of AtCAPE9, the doubly charged AtCAPE9 precursor ion (m/z 668.84) was selected with 2 Th isolation width for ion trap MS/MS full scan (m/z 180−1400) with the collision-induced dissociation (CID) energy set to 25, and product ions m/z of 294.64(b5⁺²), 325.66(y5⁺²), 344.18(b6⁺²), 536.27(y4⁺), 588.28(b5⁺), 650.31(y5⁺), 687.35(b6⁺), 749.34(y6⁺), 930.44(b8⁺), 1058.52(b9⁺) were monitored. For the MS/MS of internal standard, one doubly charged tryptic β-casein peptide (FQSpEEQQQTEDELQDK; m/z 1031.41) was selected with 2 Th isolation width for ion trap MS/MS full scan (m/z 280-1400) with the CID energy, set to 30, and product ions m/z of 747.34(y6⁺), 1105.44(y9⁺) and 1361.61(y11⁺) were monitored. The normalized abundance of AtCAPE9 was calculated by the total peak area of monitored product ions from AtCAPE9 and normalized with the abundance of the total peak areas of monitored product ions from internal standard.

To quantify the abundance of AtCAPE9 in SA-treated WT or *xcp1* plants, a linear ion trap-orbitrap mass spectrometer (Orbitrap LTQ XL, Thermo Fisher Scientific) coupled online with a nanoUHPLC system (nanoACQUITY UPLC, Waters) was used. The peptides were loaded into a 180 µm × 20-mm tunnel frit trap column[48] packed with 5 µm Symmetry C18 resin (Waters) and separated online with a 75 µm × 350-mm tunnel frit analytical column packed with 1.7 µm BEH C18 resin (Waters). Peptides were separated using an 89 min linear gradient from 4 to 25% ACN with 0.1% formic acid at 300 nL/min flow and a column temperature of 35 °C. Totally, 100 fmoles of angiotensin I peptide (#A9650, Sigma-Aldrich) was spiked before zip-tip desalting for AtCAPE9 abundance normalization. The MS was set to one full FT-MS scan (m/z 350−1650) with 60,000 resolution, followed by ion trap MS/MS full

scans for AtCAPE9 and internal standard quantification. For MS/MS of AtCAPE9, the triply charged AtCAPE9 precursor ion (m/z 446.23) was selected with 2 Th isolation width for ion trap MS/MS full scan (m/z 120–1400) with the CID energy set to 35, and product ions m/z of 294.64($b5^{+2}$), 325.66($y5^{+2}$), 344.18($b6^{+2}$), 536.27($y4^{+}$), 588.28($b5^{+}$), 650.31($y5^{+}$), 687.35($b6^{+}$), 749.34($y6^{+}$), 930.44($b8^{+}$), 1058.52($b9^{+}$) were monitored. Triple charged angiotensin I peptide (m/z 433.17) was selected with 2 Th isolation width for MS/MS full scan (m/z 115–2000) with the CID energy set to 35 and product ion m/z of 269.16($y2^{+}$), 325.67($y5^{+2}$), 392.71($b6^{+2}$), 513.28($y8^{+2}$), 583.30($b9^{+2}$), 647.35($b5^{+}$), and 784.41($b6^{+}$) were monitored. The normalized abundance of AtCAPE9 was calculated by the total peak area of monitored product ions from AtCAPE9 and normalized with the abundance of the total peak areas of monitored product ions from internal standard.

For phytohormone quantitation, a linear ion trap-orbitrap mass spectrometer (Orbitrap Elite, Thermo Fisher Scientific) coupled online with a UHPLC system (ACQUITY UPLC, Waters) was used. The phytohormones were separated by an HSS T3 column (#186004680, Waters) using gradients of 0.5–25% ACN at 0–2 min, 25–75% ACN at 2–7 min, and 75–99.5% ACN at 7–7.5 min. The mass spectrometer was operated in the negative ion mode and set to one full FT-MS scan (m/z 100–600) with 15,000 resolution, followed by two 15,000 resolution FT-MS product ion scans (m/z 50–100) using higher energy collisional dissociation (HCD) for precursor ions of SA (m/z of 137.02) and $d_4$-SA (m/z 141.05, $d_4$-SA is dissociated from the added $d_6$-SA) with 2 Th isolation width. The fragmentation reactions of m/z 137.02 to 93.03 for SA and 141.05 to 97.06 for $d_4$-SA were selected for quantitation. The absolute abundances of SA were calculated by the abundance of $d_4$-SA.

## Untargeted peptidome profiling using LC−MS/MS

To identify global peptides in SA-treated WT samples, a quadrupole-orbitrap mass spectrometer (Q Exactive HF, Thermo Fisher Scientific) coupled online with a nanoUHPLC system (nanoACQUITY UPLC, Waters) was used. The sample was loaded into and separated online with a 75 µm × 500-mm tunnel frit analytical column packed with 500 mm of 1.9 µm ReproSil-Pur 120 Å C18-AQ resin (Dr. Maisch GmbH) and was separated using a 95 min linear gradient from 8 to 40% ACN with 0.1% formic acid at 250 nL/min flow and column temperature of 40 °C. The DDA acquisition parameters were set to one full MS scan (m/z 350–1650) with 60,000 resolution and were switched to 20 product ion scans with 30,000 resolution, 28% normalized collision energy (NCE) and isolation width 1.2 Th when a precursor ion charge was 2+ to 5+ and an intensity greater than 130,000 was detected. The MS/MS spectra generated from Q Exactive HF were subjected to peak picking by msConvert version 3.0.18353 included in ProteoWizard Toolkit[49] and were output into Mascot generic format (mgf) file. The obtained mgf file was searched against a database combining the target and decoy *Arabidopsis* hypothetical peptide database without specifying enzyme cleavage rules using Mascot MS/MS ion search (Matrix Science, server version 2.3). The mass tolerance in the Mascot search for peptide precursors and fragments was set to 5 ppm and ±0.05 Da, respectively. Oxidation on Met, hydroxylation on Pro, phosphorylation on Ser/Thr, and sulfation on Tyr were specified as variable modifications. All of the MS/MS spectra identified by Mascot were subsequently rescored by calculating the delta score (DS)[13,50]. The number of peptide hits produced from decoy sequences was used to evaluate the cutoff of DS for limiting the false discovery rate (FDR) < 1% in the untargeted peptide identification result. The approach to generate the target and decoy hypothetical peptide database has been reported in our previous work[13]. In this study, the *Arabidopsis* target hypothetical peptide database was composed by extracting 50 residues of all protein C-terminal sequences from the *Arabidopsis* TAIR protein database (release version 11) plus bovine β-casein protein sequence. The *Arabidopsis* decoy hypothetical peptide database was generated by shuffling the sequences in the target databases using the algorithm option of randomizing sequences and interleave entries in Trans Proteomics Pipeline (TPP)[51].

## Assay for protease activity

To test the protease activity against different substrates, the substrate reaction buffer consists of 50 mM 4-Morpholinepropanesulfonic acid (MOPS) (#M1254, Sigma-Aldrich) with 0.1% 3-[(3-cholamidopropyl) dimethylammonio]-1-propanesulfonate (CHAPS) (#C3023, Sigma-Aldrich) at pH 6.0. To investigate the specificity of the protease, three synthetic peptides (CNYD, CNAD, and ANAD) tagged by 7-amino-4-methylcoumarin (AMC) at the C-terminus were purchased from Mission Biotech. These substrates were incubated with *Arabidopsis* protein extract or the purified XCP1-His protein in pH 6.0 reaction buffer, then the protease activity was analyzed through the detection of the fluorophore released from the substrates. The reaction sample was transferred to Optiplate™-96 F black plates (#6005270, PerkinElmer), and the fluorescent signal of the released fluorophore was detected by an ELISA microplate reader (BioTek Synergy H1) using 360 nm excitation and 460 nm emission. The enzyme activity was reported as relative fluorescent units (RFU). The RFU value was calculated by subtracting the signal of the substrate alone from the signal of the reaction sample. To evaluate the effects of metal ions on CNYDase activity, the *Arabidopsis* protein extract was incubated with 100 mM $ZnCl_2$, $MgCl_2$, and $CaCl_2$ or 5 mM EDTA in reaction buffer for 1 h, and then the CNYDase activity was measured after 5 h incubation with the substrate. The synthetic affinity aldehyde inhibitor using the CNYD sequence tagged by biotin at N-terminus and the CHO aldehyde group at C-terminus (designated as biotin−CNYD−CHO) was purchased from Biotools. To examine the effects of protease inhibitors with or without calcium activation, the *Arabidopsis* protein extract was incubated with 500 µM E-64 (#E3132, Sigma-Aldrich), 200 µM phenylmethanesulfonyl fluoride (PMSF) (#10837091001, Sigma-Aldrich) or 100 µM biotin-CNYD−CHO in reaction buffer for 1 h with or without 100 mM $CaCl_2$ and the CNYDase activity was measured after 10 h incubation with the substrate. Dose-dependent inhibition of biotin−CNYD−CHO was examined by the CNYDase activity. The *Arabidopsis* protein extract was incubated with 0, 25, 50, 100, and 200 µM biotin−CNYD−CHO for 1 h, and the CNYDase activity was measured after 10 h incubation with the substrate. The CNYDase activity of the protein extract from WT and the T-DNA insertion mutants of protease candidates were measured after 5 h incubation with the CNYD substrate. All proteolytic activity assays of the protein extract were performed by incubating 50 µg protein extract with 25 µM substrate. To study the XCP1-His activity at different pH, the substrate reaction buffer of 100 mM sodium acetate with 0.1% CHAPS in the range of pH 4.0–5.5 and 50 mM MOPS with 0.1% CHAPS in the range of pH 6.0–7.5. For the enzyme kinetics of XCP1 activity at different temperatures, the $V_{max}$ and $K_m$ of proteolytic activity were determined by 2.5 h incubation of 1 µg purified XCP1-His with different concentrations of CNYD substrate at 22, 32, or 37 °C. To determine the XCP1 activity at 22 °C, the $V_{max}$ and $K_m$ of proteolytic activity were determined by 10 h incubation of 0.2 µg purified XCP1-His with different concentrations of CNYD substrate. To examine the XCP1-His and XCP1$^{C161A}$-His of CNYDase activity, the CNYDase activity was measured after 5 h incubation of 0.5 µg purified XCP1-His or XCP1$^{C161A}$-His purified protein with 25 µM CNYD substrate reaction buffer of 50 mM MOPS with 0.1% CHAPS at pH 6.0. in the presence or absence of 400 µM biotin−CNYD−CHO at 32 °C.

## Immunoblotting and immunoprecipitation for PR1-eYFP processing

Generally, plant tissues were frozen in liquid nitrogen and ground into a fine powder, then dissolved in the plant extraction buffer (50 mM Tris-HCl, pH 7.5, 150 mM NaCl, 20 mM EDTA, 1% SDS, and 10% glycerol)

and rotated at 50 rpm for 30 min at 4 °C. The protein extract was then centrifuged at 16,000 × g at 4 °C for 15 min, and the supernatant was filtered by 70 μm Nylon cell strainers (Falcon). The protein concentration was measured, and 50 μg proteins were dissolved by 5× sample buffer (250 mM Tris at pH 6.8 with 50% glycerol, 5% SDS, 0.02% bromophenol blue and 5% β-mercaptoethanol) for heating at 95 °C for 5 min. The proteins were separated by 12.5% SDS-PAGE gels (1.5 mm) and transferred onto PVDF membranes (#10600023, GE Healthcare) by wet western blotting. Membranes were first blocked with 5% milk in Tris buffer saline-Tween 20 (TBST) buffer and then incubated with primary antibodies at 4 °C for overnight. Protein complexes were labeled with HRP-conjugated secondary antibodies (mouse, 1:5000; # 61-6520, Thermo Fisher Scientific) at room temperature for 1 h. All HRP-conjugated proteins were detected on the membrane with enhanced chemiluminescence (ECL) reagent kit (#193508, Biotools) reacting to the HRPs. For the analysis of PR1-eYFP processing, eYFP-tagged proteins were recognized by anti-GFP antibody (mouse, 1:5000; #11814460001, Roche) For confirming the protein expression of different forms of PR1 in WT and *pr1*, proteins were recognized by anti-PR1 antibody (Rabbit, 1:2500, #AS10687, Agrisera). For confirming the protein expression and amount of recombinant proteins tagged with C-terminal 6×His, proteins were recognized by anti-His antibody (mouse, 1:5000; #PPT-66005-1, Biotools) For detecting biotin−CNYD−CHO-bounded proteins, proteins were recognized by streptavidin-HRP (1:5000; #016-030-084, Jackson ImmunoResearch).

For prove the PR1-eYFP is processed by the purified native or mutated XCP1-His, the immunoprecipitation of PR1-eYFP was performed by anti-PR1 antibody (Rabbit, 1:2500, #AS10687, Agrisera) immobilized on the Pierce™ Protein A/G Magnetic Beads (#88803, Thermo Fisher Scientific) and the immobilized PR1-eYFP proteins were incubated with 1.0 μg purified native or mutated XCP1-His protein in 50 mM MOPS with 0.1% CHAPS at pH 6.0 for 1 h. The elution of the intact PR1-eYFP and the proteolytic fragment of AtCAPE9-eYFP using IgG Elution Buffer, pH 2.0 (#21028, Thermo Fisher Scientific) from the Protein A/G Magnetic Beads was detected by anti-GFP antibody (mouse, 1:1000; #11814460001, Roche).

### Reporting summary
Further information on research design is available in the Nature Portfolio Reporting Summary linked to this article.

## Data availability
The data that support the findings of this study are available within this manuscript and its Supplementary Information and Supplementary Data file. MS raw data for peptidomics analysis were deposited to the ProteomeXchange Consortium via the PRIDE partner repository with the dataset identifier PXD041898. MS raw data files for targeted MS analysis using the PRM method were deposited to the MassIVE with dataset identifier MSV000091833 [https://doi.org/10.25345/C5K35MQ31]. All materials, including plant lines, bacteria strains, plasmids, and primers used in this study, were described in the Supplementary Information. All the unprocessed data, gels, and blots were provided in the Supplementary/Source Data file. Data or material used in this paper is also available from the corresponding author upon request. Source data are provided in this paper.

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

## Acknowledgements

This work was financially supported by the Academia Sinica (Career Development Award, CDA-105-L03) and the Ministry of Science and Technology of Taiwan (MOST 107-2113-M-001-006). The MS analysis was supported by the Metabolomics Facilities of the Scientific Instrument Center at Academia Sinica (AS-CFII-111-218). We thank Plant Tech. Core Laboratory, Agricultural Biotechnology Research Center, Academia Sinica for CRISPR/Cas9 system support. We thank Life Science Editors for editorial assistance.

## Author contributions

The authors employed by Academia Sinica conducted all of the experiments presented in this manuscript. Y.-R.C. conceptualized/supervised the project, acquired all funding, designed experiments for identifying XCP1 as ESCAPE and the role of PR1/AtCAPE9/XCP1 for SAR.; F.-W.L. characterized the XCP1 as the protease for AtCAPE9, demonstrate the XCP1-PR1 interaction, studied the CNYDase activity in plants, purified XCP1, and performed most of the experiments.; F.-W.L. and K.-T.C. demonstrated the production of AtCAPE9 from PR1 by XCP1 is essential for SAR.; Y.-L.C. performed screening of protease candidate, identified endogenous AtCAPE9 and demonstrate its role on SA induction. F.-W.L. and K.-T.C. prepared and verified all the transgenic plants.; C.-H.C designed and generated pr1 mutant using CRISPR/Cas9 system and performed the subcellular localization of XCP1-GFP and PR1-eYFP.; S.-C.H. quantified the abundance of endogenous AtCAPE9 in WT and xcp1 mutant.; K.-T.C. operated all pathogen experiments and recorded the data.; T.E. performed the molecular docking analysis to identify enzyme candidates.; Y.-L.C., C.-H.C., and S.-C.H. analyzed the data.; Y.-L.C. and C.-H.C. contributed ideas and designed specific experiments.; C.-H.C. and S.-C.H. validated and prepared all the data for publication.; Y.-R.C., Y.-L.C., C.-H.C., and S.-C.H. wrote and revised the manuscript with help from all of the co-authors.

## Competing interests

The authors declare no competing interests.
