## [Peer Review File · Nature Communications]

XCP1 cleaves Pathogenesis-related protein 1 into CAPE9 for systemic immunity in ArabidopsisReviewer #1 (Remarks to the Author):

This is a very interesting manuscript showing the role of XCP1 in the release of the CAPE9 signaling peptide from its precursor, thereby mediating SAR in systemic leaves. The quality of the work is excellent and the science is sound. The conclusions are supported by the data. There are several important issues and some minor comment.

- 1) Please do not rename XCP1 into ESCAPE. XCP1 is well known in the research community and renaming this gene will be very confusing and is not necessary.
- 2) Please include the XCP1C161A as a control in the PR1 processing experiments (Fig5fg).
- 3) The focus to XCP1 (L141-150) should be rephrased. It is unclear why XCP2 was excluded. Perhaps a better way to reason is that: 1) XCPs are localised in the xylem; 2) they are expressed and detected; 3) of the 9 tested PLCP mutants, only xcp1 mutant has less CNYDase activity (FigS10). Please also discuss that the residual CNYDase activity and SAR might be caused by XCP2.
- 4) I am puzzled by the fact that an aldehyde probe (Biotin-CNYD-CHO) was used for covalent labelling. CMK/FMK are covalent probes, but CHO not. Please add a reference to support covalent labelling by aldehyde probes.
- 5) L196 The secretion of XCP1 and PR1 by the presented cell biology experiments are not that convincing since they are overexpressed, but there is a lot of literature on the detection of XCP1/PR1 in the apoplast/xylem that should be cited.
- 6) Presentation of % cleavage data in Log2FC is confusing in Fig3b and Fig6c. It is probably better to show the original data.

Minor issues:

- L50 also VPE and PBE1 have caspase activity.
- L81 Delete 'we overexpressed a series...Specifically'. Three AtPR1s are not 'a series.'
- L90: Given that this is all Arabidopsis work, please use PR1 and pr1 throughout, rather than AtPR1 and atrp1.
- L91 Why had the pr1 mutant to be generated by genome editing? Are there no T-DNA mutants described/available?
- L99 'transfection' is confusing. Better would be 'transient expression upon incubation of seedling with Agrobacteria.'
- L110 What are 'fragmentation transitions'? Please rephrase.
- L143 Why are metacaspases considered if they are not sensitive to E64?
- L184 Refs 19,20 do not describe CNYDase activity. Please check references throughout as some might have shifted.
- L186 'dramatically lower' perhaps better is 'no CNYDase activity was detected'
- L231 Please check the sentence composition
- L239 XCP2 does not aid in TE formation.
- L245-247 Citation for Avr2 is missing. Cladosporium fulvum is not a pathogen of Arabidopsis, so this is highly speculative. This section might need an introductory sentence that the observations on Arabidopsis XCP1-CAPE signalling might also occur in other plants. In that context, the Avr2 and FolsVp1 effectors make more sense.
- L285 Can you detect reduced CAPE9 release at higher temperatures? This would be an interesting experiment to do, rather than to speculate on.
- L288 Please discuss how CAPE8 may signal SAR in concert with other SAR signals.
- L288-289 Please check the sentence composition.
- L448 Nicotiana benthamiana is a wild relative of tobacco and not tobacco.
- L518 Please check the sentence composition. The sentence is vague.
- L584-585 Did the authors incubate 12-day old At seedlings in bacterial solution for 7 days?
- L985 In the legends, authors refer to tobacco which I presume is N. benthamiana.
- L985 Please add a reference for PIP2 as PM maker.
- Fig4 Were these Arabidopsis plants all untreated?

Reviewer #2 (Remarks to the Author):

In the manuscript entitled "XCP1 cleaves Pathogenesis-related protein 1 into CAPE9 for systemic immunity in Arabidopsis", Chen et al., described the identification and characterisation of the caspase involved in the generation of the cytokine AtCAPE9 from Arabidopsis Pathogenesis-Related Protein 1 (AtPR1). The authors found that the C-terminal proteolytic processing of caspase-like substrate motif 'CNYD' in AtPR1 produces AtCAPE9. Treatment with salicylic acid (SA) or its analog INA enhanced the CNYD-targeted protease activity and the generation of AtCAPE9 from AtPR1. The cleavage of substrate motif is observed when the substrate is native (CNYDP) but not an alanine-substituted version of the motif (CNYAP and CNYDA). The CNYDase assays showed that cleavage of the Ac-CNYD-AMC substrate was enhanced by CaCl₂ but inhibited by ZnCl₂ and EDTA. The Ca²⁺-enhanced CNYDase activity was suppressed by the general cysteine protease inhibitor E-64. Furthermore, the authors detected the biotin-CNYD-CHO-binding proteins by immunoblot analysis, suggesting that a potential CNYDase has a MW of 35 kDa. These biochemical results suggest that the protease should be 35 kDa and Ca²⁺-activated cysteine-dependent/aspartate-specific CNYDase. To identify the candidate protease, the authors analysed 1) the molecular weight of pro-form/processed form and 2) the expression patterns of cysteine protease families and identified XYLEM CYSTEINE PEPTIDASE 1 (XCP1) as a candidate for the enzyme specific for CAPE production, which they named ESCAPE. The recombinant protein of XCP1 has Ca²⁺-dependent proteolytic activity to biotin-CNYD-CHO. The CNYDase activity was reduced in systemic leaves of *xcp1* when treated with flg22 at local leaves. The knockout mutant *xcp1* showed susceptibility against *Pseudomonas syringae* pv. tomato (Pst) DC3000 in local tissues. Furthermore, *xcp1* did not show flg22-induced systemic resistance against Pst DC3000, while the over-expression of XCP1-His in *xcp1* did. Together, the authors concluded that XCP1 is a CNYDase that generates the cytokine AtCAPE9 from AtPR1 to regulates systemic-acquired immunity.

The strengths of the study include the novelty regarding how to generate cytokine AtCAPE9 from AtPR1, a common marker gene for immunity, and how to induce AtCAPE9-mediated systemic resistance against a bacterial pathogen in Arabidopsis. The mechanism would be of interest to the readers and community of the plant immunity field. The authors performed multiple experiments to substantiate their conclusions, and most of the data are clear and convincing. However, I have several critical concerns mentioned below to help the authors improving their manuscript.

Major points

1. The authors claim that C-terminal proteolytic cleavage of AtPR1 by XCP1 generates the cytokine AtCAPE9 and induces systemic immunity. However, the mechanism of how the XCP1-cleaved AtCAPE9 peptide induces systemic immunity needs to be clarified. For instance, where the XCP1 is activated and cleaves the 'CNYD' motif in AtPR1 to produce AtCAPE9 during pathogen infection? Some previous studies showed a part of the mechanism: Pečenková et al., *Mol Plant Pathol* 2022 reported that only intact or C-terminal motif processing-mimicking variants of AtPR1 are secreted to the apoplast when AtPR1 is transiently expressed in *N. benthamiana* leaves. Baena et al., *Plant Physiol* 2022 showed that AtPR1 is secreted to the apoplast after 48h of Pst DC3000 infection in Arabidopsis. In Fig. S11, the authors also showed that XCP1-GFP and AtPR1-eYFP localised to the apoplast when both are transiently expressed in *N. benthamiana* leaves. Do the authors think that AtPR1 processing by XCP1 happens in the apoplast or intracellularly (e.g. in the secretory system) during Pst DC3000 infection?
2. In addition to the timing of the AtPR1 processing by XCP1, my other question is whether AtCAPE9 itself move systemically to induce systemic acquired resistance (SAR), or does AtCAPE9 activate SA signalling pathway at local tissue to induce SAR? The authors could clarify the mechanism by checking CAPE9-induced resistance against Pst DC3000 in SA-biosynthesis, SA-signalling, or SAR mutants.
3. The authors found that the 'CYND' motif in AtPR1 was cleaved by XCP1. A previous study by the authors (Chen et al., *Plant Cell* 2014) indicated that the 'CYND' motif is conserved not only in CAPE9/PR1 but also in CAPE1-7. Do the authors think XCP1 is also involved in the processing of these other cytokines? If not, how could specificity be determined? Does a cape9 mutant phenocopy *xcp1*, or does one need to mutate additional CAPEs?
4. The authors found that SA or INA treatment enhanced the CNYDase activity of XCP1, and the AtCAPE9 was released. On the other hand, treatment with AtCAPE9 enhanced SA content and

induced SAR against Pst DC3000. During the infection by Pst DC3000, which process happens first? Does infection of Pst DC3000 enhance the CNYDase activity of XCP1 and activate the production of CAPE9 and subsequent SA accumulation? Or does infection of Pst DC3000 induce SA accumulation, which then enhances the CNYDase activity of XCP1 and production of AtCAPE9, leading to much higher SA accumulation?

5. The characterisation of AtPR1 variants, particularly in the context of SAR induction (Fig. 2) would be preferable in stable transgenic lines, rather than after Agrobacterium-mediated transient expression.

6. From my understanding the infection assays in Fig. 6 are done with individual F2 plants – this is not sufficient to draw conclusions. These assays need to be repeated at least 3 times with homozygous T3 plants.

7. It is also puzzling that infection assays with Pst DC3000 are scored at 5 dpi or even 7 dpi, which is certainly not standard in the field. Also, why using seedlings in Fig. 1a but leaves from adult plants in other figures for such assays?

8. Regarding the cleavage assays, 10 h seems like a very long incubation time (e.g. Figs. 3 and 4). There also seems to be considerable variation between experiments in terms of RFU. Please comment.

9. Is XCP1 also required for SAR induced by an ETOI-inducing PST DC3000 strain (e.g. AvrRpm1), as most commonly used in the study of SAR in Arabidopsis?

10. Please retain the name XCP1 in the text, as it is confusing to rename genes/proteins that were already named before. Please only refer to the protease as ESCAPE before it is identified.

Minor points

- Lines 35, 56, 287: "PR1" should be "AtPR1".
- Line 67: In the final paragraph of the Introduction, the authors should write the main claim/message of the manuscript.
- Lines 141-150: the explanation of how the authors narrowed down the list of Arabidopsis papain-like cysteine protease candidates is unclear. The authors claim that they chose the protease showing 1) 35-kDa molecular weight and 2) similar expression patterns with AtPR1. It is hard to choose the target proteases based on the molecular weight by immunoblotting because the MWs of all the proteases listed by authors only differ by a few kDa. Furthermore, the expression patterns of XCP2 (AT1G20850), At3g49340, XBCP3 (At1g09850), At3g19400, At4g19390, At2g27420 and CEP3 (AT3G48350) are similar to XCP1 and AtPR1. However, the authors said XCP1 is the only candidate that fits the criteria. Did the authors have any other evidence that XCP1 is the candidate protease? Did the authors perform any functional screening of the proteases to narrow down the candidates?
- Lines 144-145: Also, why mentioning MCs initially as focus of investigation as they anyway do not fulfil the criteria?
- Line 191, 863: "anti-PR1 antibody" should be "anti-AtPR1 antibody".
- Fig. 2c, 3a, 4b, 5c, S6: Additional label is written behind the label. Please remove it.
- Fig. 3b, 6c, S10: The vertical axis is not an appropriate scale. Please revise it.
- Fig. 6a, 6b, 6e, 6f, S1a, S1d, S3, S5, S7c: Line frames are displayed because of the backfill.
- Fig. S5d: The authors should perform statistical analysis for the data.
- Please could the authors quantify the correlation between the spectra in Fig. 1b.
- Please plot bar charts so that individual datapoints are visible.
- What do the authors propose the middle band is in Fig. S2 (and other later blots)? Why does this not appear in Fig. 1c? Please could the authors treat the plants in Fig. 1c with SA/INA to see whether this product is still formed under these conditions.
- Please define Ac-CNYD-AMC the first time it is used.
- Do the authors have a hypothesis as to why mutating most of the cysteine proteases in S6 leads to significant enhancement of cleavage of the CNYD reporter?
- Could the endogenous CAPE9 levels be normalised to an endogenous peptide from a housekeeping protein?

- Fig. 5b should have a loading control
- Please change the nomenclature for the crosses – currently it makes me think the lines are F1s rather than being homozygous.
- Refs 11 and 13 are the same.
- Throughout the manuscript: as all forms of plant immunity are innate, it does not make sense to refer to plant innate immunity.
- Line 244: precise that these findings were in tomato.

References:

Baena, G., Xia, L., Waghmare, S., & Karnik, R. (2022). SNARE SYP132 mediates divergent traffic of plasma membrane H⁺-ATPase AHA1 and antimicrobial PR1 during bacterial pathogenesis. *Plant physiology*, 189(3), 1639–1661. <https://doi.org/10.1093/plphys/kiac149>

Pečenková, T., Pejchar, P., Moravec, T., Drs, M., Haluška, S., Šantrůček, J., Potocká, A., Žárský, V., & Potocký, M. (2022). Immunity functions of Arabidopsis pathogenesis-related 1 are coupled but not confined to its C-terminus processing and trafficking. *Molecular Plant Pathology*, 23(5), 664–678. <https://doi.org/10.1111/mpp.13187>

Chen, Y. L., Lee, C. Y., Cheng, K. T., Chang, W. H., Huang, R. N., Nam, H. G., & Chen, Y. R. (2014). Quantitative peptidomics study reveals that a wound-induced peptide from PR-1 regulates immune signaling in tomato. *The Plant Cell*, 26(10), 4135–4148. <https://doi.org/10.1105/tpc.114.131185>

Responses to the Reviewers

We are grateful for the constructive suggestions and comments made by reviewers. In addition, we appreciate the time and effort invested by the reviewers in evaluating our manuscript. We are confident that the revised manuscript will be a valuable addition to the literature in the plant immunity field. Below are our point-by-point responses to the reviewers' comments. We also uploaded a copy of the manuscript text highlighting the changes as an additional supplementary file for your reference.

Reviewer #1 (Remarks to the Author):

This is a very interesting manuscript showing the role of XCP1 in the release of the CAPE9 signaling peptide from its precursor, thereby mediating SAR in systemic leaves. The quality of the work is excellent and the science is sound. The conclusions are supported by the data. There are several important issues and some minor comment.

We are glad the reviewer found our work interesting and of high quality. All of the major and minor comments from the reviewer have been taken into careful consideration and addressed in the revised manuscript. We appreciate the reviewer's thoughtful comments and outstanding suggestions, which have helped us to improve the clarity of the manuscript and enhance the impact of our work. We are confident these revisions have significantly strengthened the manuscript and its contribution to the field.

Comment 1-1

Please do not rename XCP1 into ESCAPE. XCP1 is well known in the research community and renaming this gene will be very confusing and is not necessary.

Response:

To avoid confusion within the research community, the gene name of *XCP1* in the revised manuscript has been retained. As suggested by reviewer 2, we refer to the term "enzyme specific for CAPE (ESCAPE)" only when discussing its role within the plant and in the identification process.

Comment 1-2

Please include the XCP1C161A as a control in the PR1 processing experiments (Fig5fg).

Response:

We have addressed the reviewer's suggestion by including His-tagged XCP1(C161A) as an additional control to reperform the PR1 processing experiments for Fig. 5f, g. In the revised Fig. 5g, we used gentler shaking during the interaction of XCP1-His with immobilized PR1-eYFP to minimize the

detection of abundant heavy chain from antibody and PR1-eYFP that may have dissociated from the immobilization particles.

Comment 1-3

The focus to XCP1 (L141-150) should be rephrased. It is unclear why XCP2 was excluded. Perhaps a better way to reason is that: 1) XCPs are localised in the xylem; 2) they are expressed and detected; 3) of the 9 tested PLCP mutants, only xcp1 mutant has less CNYDase activity (FigS10) (we think the reviewer means FigS6 in the original manuscript). Please also discuss that the residual CNYDase activity and SAR might be caused by XCP2.

Response:

We sincerely thank the suggestion from the reviewer for clarifying how XCP2 can be excluded from the candidates. The data for 9 tested PLCP mutants (Fig. S6 in the original manuscript) has been included in Supplementary Fig. 6 in the revised manuscript. Line 141-150 in the original manuscript has been rephrased based on the reviewer's suggestion (Line 161-173 in the revised manuscript). We also discuss the potential involvement of XCP2 or other proteases in CNYDase and SAR (Line 287-289 in the revised manuscript).

Comment 1-4

I am puzzled by the fact that an aldehyde probe (Biotin-CNYD-CHO) was used for covalent labelling. CMK/FMK are covalent probes, but CHO not. Please add a reference to support covalent labelling by aldehyde probes.

Response:

We have found a study indicating that the peptide aldehyde protease inhibitor labels cysteine proteases via forming covalent hemithioacetal adducts that mimic the tetrahedral transition states of the enzyme reaction pathway in the catalytic center. Unlike CMK/FMK-derived protease inhibitors that tag the Cys residue in the catalytic center via an irreversible covalent bonding, the aldehyde probe can form strong but reversible covalent bonding with the Cys. This may be why some literature described the aldehyde inhibitor as non-covalent. To clarify the reason for using this aldehyde probe, a study showing that the aldehyde probe labels the Cys covalently has been added to the revised manuscript¹ (Line 146-148 in the revised manuscript).

Comment 1-5

L196 The secretion of XCP1 and PR1 by the presented cell biology experiments are not that convincing since they are overexpressed, but there is a lot of literature on the detection of XCP1/PR1 in the apoplast/xylem that should be cited.

Response:

We have added several references to support the secretion of PR1 and XCP1 in the apoplast/xylem (Line 215-217 in the revised manuscript).

Comment 1-6

Presentation of % cleavage data in Log2FC is confusing in Fig3b and Fig6c. It is probably better to show the original data.

Response:

One of the gel images for Fig. 3b or Fig. 6c from Supplementary Fig. 3 or Supplementary Fig. 14 has been respectively added to the main figure to provide a more straightforward representation of the data.

Minor Issue/Point 1-1

L50 also VPE and PBE1 have caspase activity.

Response:

We appreciate the reviewer pointing out two important caspase-like enzymes that we overlooked in the introduction: the vacuolar processing enzyme (VPE) and a subunit from the proteasome. We found that the proteasome subunit PBA1 has been shown to have caspase activity, but there has not yet been reported for PBE1. Therefore, we have added VPE and PBA1 to the introduction section of the manuscript to provide a more comprehensive overview of caspase-like enzymes in plants (Line 48-53 in the revised manuscript).

Minor Issue/Point 1-2

L81 Delete 'we overexpressed a series...Specifically'. Three AtPR1s are not 'a series.'

Response:

The sentence has been modified to "we individually overexpressed different sequences of PR1 fused to enhanced yellow fluorescent protein (eYFP) in *Arabidopsis*." (Line 91-92 in the revised manuscript).

Minor Issue/Point 1-3

L90: Given that this is all Arabidopsis work, please use PR1 and pr1 throughout, rather than AtPR1 and atpr1.

Response:

All of the AtPR1, *AtPR1* and *atpr1* throughout the original manuscript have been changed to PR1, *PR1* and *pr1*, respectively.

Minor Issue/Point 1-4

L91 Why had the *pr1* mutant to be generated by genome editing? Are there no T-DNA mutants described/available?

Response:

The reviewer pointed out a critical issue regarding the study of PR1 using T-DNA *pr1* mutant. To the best of our knowledge, there is no reliable T-DNA insertion *PR1* knockout (KO) mutant in the database, as this issue has been previously mentioned². We attempted to test available T-DNA insertion *pr1* insertion mutant lines (SAIL_440_A12 and GK-472B03) from the Arabidopsis Biological Resource Center (ABRC). Although for the SAIL_440_A12 line, the ABRC reported that the T-DNA is inserted in the coding region of *PR1*, neither this genotype nor *PR1* KO expression in SAIL_440_A12 was detected in our study (we tested several batches of these lines delivered by ABRC, as shown in the figure below). For the GK-472B03 line, the T-DNA has been reported to be inserted in the promoter region of *PR1*. We could detect the expected insertion genotype and *PR1* knockdown expression in this line. However, we occasionally observed *PR1* expression in this line, which was originally repressed at the seedling stage but turned overexpressed in the adult stage (and *vice versa*) even though this line has been determined to be a homozygous mutant (as shown in the figure below). This is why we spent more than three years obtaining a *PR1* CRISPR mutant. Another group also utilized *PR1* CRISPR mutant³, probably due to the same issue.

Figure. Genotyping and expression of *PR1* gene in T-DNA insertion *pr1* mutant lines (GK-472B03 and SAIL_440_A12).

(a) Genotyping of *PR1* gene in WT and SAIL_440_A12. (b) *PR1* expression in WT and SAIL_440_A12. (c) Genotyping of *PR1* gene in WT and GK-472B03. (d) *PR1* expression in WT and GK-472B03. R1 and R2 indicate individual experiments using GK-472B03 homozygous lines at different growth stages. *ACTIN2* was used as internal control.

Minor Issue/Point 1-5

L99 'Transfection' is confusing. Better would be 'transient expression upon incubation of seedling with Agrobacteria.'

Response:

We appreciate the suggestion to provide a clearer description of the experiment. The sentences "transfection of P19 plus native AtPR1....." in Line 99 and "transfection of P19, P19 plus AtPR1^{D150A} or P19 plus AtPR1^{ΔCAPE9}" in Line 101 of the original manuscript have been revised to "transient expression of P19 plus native PR1" and "transient expression of P19, P19 plus PR1^{D150A} or P19 plus PR1^{ΔCAPE9}", respectively (Line 109-112 in the revised manuscript).

Minor Issue/Point 1-6

L110 What are 'fragmentation transitions'? Please rephrase.

Response:

The sentence "Briefly, LC-MS/MS was operated in multiple reaction monitoring (MRM) mode targeting the fragmentation transitions of AtCAPE9." has been rephrased to "Briefly, LC-MS was operated in targeted MS/MS mode by selecting the precursor and specific fragment ions of AtCAPE9." (Line 119-120 in the revised manuscript).

Minor Issue/Point 1-7

L143 Why are metacaspases considered if they are not sensitive to E64?

Response:

Excluding most of the MCs by screening the activity using E-64 is a good idea since several MCs are not sensitive to E-64. We recently came across a report showing that E-64 can only partially inhibit MC4 at a concentration of 100 μM⁴. According to the inhibition assay performed in Fig. 4, we may not be able to exclude all MCs, such as MC4, using a high concentration of E-64 (500 μM) in the assay. To ensure a complete exclusion of all MCs, applying a lower concentration of E-64 for typing enzyme activity would be advisable.

Minor Issue/Point 1-8

L184 Refs 19,20 do not describe CNYDase activity. Please check references throughout as some might have shifted.

Response:

We thank the reviewer pointed out some shifted citations. References 19 and 20 in the original manuscript have been moved to the correct locations (References 20 and 21 in Line 203 in the revised

manuscript). We have also checked all the other references for their locations and fixed the shifted ones (Reference 29 in Line 272-273 in the revised manuscript).

Minor Issue/Point 1-9

L186 ‘dramatically lower’ perhaps better is ‘no CNYDase activity was detected’

Response:

The sentence has been modified to “Indeed, no CNYDase activity was detected from XCP1^{C161A}-His.” (Line 205-206 in the revised manuscript).

Minor Issue/Point 1-10

L231 Please check the sentence composition

Response:

The sentence has been modified to “Further, the treatment of AtCAPE9 to the local leaves of *pr1* directly enables the systemic immunity without *Agrobacterium* infection, which implies AtCAPE9 may directly or indirectly elicit systemic immunity.” (Line 257-259 in the revised manuscript).

Minor Issue/Point 1-11

L239 XCP2 does not aid in TE formation.

Response:

The XCP2 has been removed from the sentences (Line 265 in the revised manuscript).

Minor Issue/Point 1-12

L245-247 Citation for Avr2 is missing. *Cladosporium fulvum* is not a pathogen of *Arabidopsis*, so this is highly speculative. This section might need an introductory sentence that the observations on *Arabidopsis* XCP1-CAPE signalling might also occur in other plants. In that context, the Avr2 and FolSvp1 effectors make more sense.

Response:

We thank the advice from the reviewer. We have added an introductory sentence to the paragraph stating that the observations on *Arabidopsis* XCP1-CAPE signaling might also occur in other plants, or the infection by other pathogens. We have also included a citation for Avr2 and modified the sentence to clarify that Avr2-expressing plants exhibiting higher susceptibility may be due to the disruption of XCP1's CNYDase activity, leading to reduced production of AtCAPE9 and SA. Additionally, we have emphasized that the mechanism by which FolSvp1 suppresses CAPE production may also reduce tomato resistance to *Fusarium oxysporum*. Finally, we have noted that PR1 and XCP1 are highly conserved in the plant kingdom, suggesting that this mechanism may also be observed in other plant

systems (Line 271-275 and Line 283-284 in the revised manuscript).

Minor Issue/Point 1-13

L285 Can you detect reduced CAPE9 release at higher temperatures? This would be an interesting experiment to do, rather than to speculate on.

Response:

We acknowledge the suggestion for this interesting and critical study. An additional experiment has been performed and found that the induction of PR1-eYFP cleavage into AtCAPE9-eYFP by INA in *Arabidopsis* is actually reduced at 32°C as compared to 22°C (Supplementary Fig. 10).

Minor Issue/Point 1-14

L288 Please discuss how CAPE8 (we think the reviewer means CAPE9 here) may signal SAR in concert with other SAR signals.

Response:

We have added a proposed pathway for flg22-elicited systemic immunity and discuss how SA or Pip/NHP may be involved in this process (Fig. 7, Line 317-328 in the revised manuscript).

Minor Issue/Point 1-15

L288-289 Please check the sentence composition.

Response:

The sentence has been revised to "Analogous to animal caspases that activate and secrete immunomodulatory cytokines, plants can utilize a caspase-like protease XCP1 to target and proteolyze the PR1 as pro-cytokine for releasing the phyto cytokine AtCAPE9 for systemic immunity." (Line 341-343 in the revised manuscript).

Minor Issue/Point 1-16

L448 *Nicotiana benthamiana* is a wild relative of tobacco and not tobacco.

Response:

We thank the reviewer for this correction. The *Nicotiana benthamiana* is the only species used for protein expression in this manuscript. All the term "tobacco" used in the manuscript has been corrected to *Nicotiana benthamiana*.

Minor Issue/Point 1-17

L518 Please check the sentence composition. The sentence is vague.

Response:

The sentence has been revised to "To generate the *XCP1* complementation lines, we expressed the *XCP1-His* gene driven by the native promoter in the T3 generation of the *xcp1* mutant." (Line 583-584 in the revised manuscript).

Minor Issue/Point 1-18

L584-585 Did the authors incubate 12-day old At seedlings in bacterial solution for 7 days?

Response:

Yes, the seedlings were actually dipped in bacterial solution for 7 days.

Minor Issue/Point 1-19

L985 In the legends, authors refer to tobacco which I presume is *N. benthamiana*.

Response:

It is actually *N. benthamiana*, the "tobacco" has been corrected to *N. benthamiana*.

Minor Issue/Point 1-20

L985 Please add a reference for PIP2 as PM marker.

Response:

A reference for PIP2A as the PM marker for Supplementary Fig. 12 has been cited in the manuscript (Line 220-221 in the revised manuscript).

Minor Issue/Point 1-21

Fig4 Were these Arabidopsis plants all untreated?

Response:

We have modified the sentence "Each assay was performed by incubating 50 µg protein extract from *Arabidopsis* with 25 µM substrate." in Fig. 4 to "Each assay was performed by incubating 50 µg protein extract from untreated *Arabidopsis* with 25 µM substrate."

Reviewer #2 (Remarks to the Author):

In the manuscript entitled “XCP1 cleaves Pathogenesis-related protein 1 into CAPE9 for systemic immunity in Arabidopsis”, Chen et al., described the identification and characterisation of the caspase involved in the generation of the cytokine AtCAPE9 from Arabidopsis Pathogenesis-Related Protein 1 (AtPR1). The authors found that the C-terminal proteolytic processing of caspase-like substrate motif ‘CNYD’ in AtPR1 produces AtCAPE9. Treatment with salicylic acid (SA) or its analog INA enhanced the CNYD-targeted protease activity and the generation of AtCAPE9 from AtPR1. The cleavage of substrate motif is observed when the substrate is native (CNYDP) but not an alanine-substituted version of the motif (CNYAP and CNYDA). The CNYDase assays showed that cleavage of the Ac-CNYD-AMC substrate was enhanced by CaCl₂ but inhibited by ZnCl₂ and EDTA. The Ca²⁺-enhanced CNYDase activity was suppressed by the general cysteine protease inhibitor E-64. Furthermore, the authors detected the biotin-CNYD-CHO-binding proteins by immunoblot analysis, suggesting that a potential CNYDase has a MW of 35 kDa. These biochemical results suggest that the protease should be 35 kDa and Ca²⁺-activated cysteine-dependent/aspartate-specific CNYDase. To identify the candidate protease, the authors analysed 1) the molecular weight of pro-form/processed form and 2) the expression patterns of cysteine protease families and identified XYLEM CYSTEINE PEPTIDASE 1 (XCP1) as a candidate for the enzyme specific for CAPE production, which they named ESCAPE. The recombinant protein of XCP1 has Ca²⁺-dependent proteolytic activity to biotin-CNYD-CHO. The CNYDase activity was reduced in systemic leaves of *xcp1* when treated with flg22 at local leaves. The knockout mutant *xcp1* showed susceptibility against *Pseudomonas syringae* pv. tomato (Pst) DC3000 in local tissues. Furthermore, *xcp1* did not show flg22-induced systemic resistance against Pst DC3000, while the over-expression of XCP1-His in *xcp1* did. Together, the authors concluded that XCP1 is a CNYDase that generates the cytokine AtCAPE9 from AtPR1 to regulates systemic-acquired immunity.

The strengths of the study include the novelty regarding how to generate cytokine AtCAPE9 from AtPR1, a common maker gene for immunity, and how to induce AtCAPE9-mediated systemic resistance against a bacterial pathogen in Arabidopsis. The mechanism would be of interest to the readers and community of the plant immunity field. The authors performed multiple experiments to substantiate their conclusions, and most of the data are clear and convincing. However, I have several critical concerns mentioned below to help the authors improving their manuscript.

We thank the reviewer for recognizing the novel strengths of our study and most of the experiments that we conducted to substantiate our conclusions. We would like to express our gratitude for the constructive feedback and outstanding questions provided by the reviewer. We have carefully

considered all comments and suggestions raised by the reviewer in revising the manuscript. We agree that addressing all the points raised by the reviewer has helped strengthen our study and contribute to advancing the field.

Major points

Comment 2-1

The authors claim that C-terminal proteolytic cleavage of AtPR1 by XCP1 generates the cytokine AtCAPE9 and induces systemic immunity. However, the mechanism of how the XCP1-cleaved AtCAPE9 peptide induces systemic immunity needs to be clarified. For instance, where the XCP1 is activated and cleaves the 'CNYD' motif in AtPR1 to produce AtCAPE9 during pathogen infection? Some previous studies showed a part of the mechanism: Pečenková et al., *Mol Plant Pathol* 2022 reported that only intact or C-terminal motif processing-mimicking variants of AtPR1 are secreted to the apoplast when AtPR1 is transiently expressed in *N. benthamiana* leaves. Baena et al., *Plant Physiol* 2022 showed that AtPR1 is secreted to the apoplast after 48h of *Pst* DC3000 infection in *Arabidopsis*. In Fig. S11, the authors also showed that XCP1-GFP and AtPR1-eYFP localized to the apoplast when both are transiently expressed in *N. benthamiana* leaves. Do the authors think that AtPR1 processing by XCP1 happens in the apoplast or intracellularly (e.g. in the secretory system) during *Pst* DC3000 infection?

Response:

The reviewer raised an outstanding question about where the XCP1 cleaves PR1 to produce AtCAPE9. In producing AtCAPE9-eYFP from PR1-eYFP, the AtCAPE9-eYFP should not be easily secreted if it is produced in the cell due to the lack of the signal peptide and a suitable transporter for AtCAPE9 fused eYFP. We could observe AtCAPE9-eYFP largely produced in the apoplast when the PR1-eYFP was highly cleaved into AtCAPE9-eYFP in the total protein extract. However, it is still difficult to exclude the possibility that the production of AtCAPE9-eYFP is in the secretory system.

Figure. Immunoblot of PR1-eYFP and AtCAPE9-eYFP in apoplastic (lane A) and total protein (lane T) extracts from transgenic *PR1-eYFP Arabidopsis*.

The sizes of the intact/uncleaved PR1-eYFP and the putative AtCAPE9-eYFP fragment were estimated to be ~44.7 and ~27.0 kDa, respectively, detected by immunoblot using anti-

GFP. Coomassie Blue staining shows total protein loaded.

In addition, without the addition of excessing Ca^{2+} , the cytosolic pH (7.0) exhibits the lowest XCP1 CNYDase activity (Fig. 5d). Compared to the cytosolic $[\text{Ca}^{2+}]$ is normally in the range of 1-200 nM, the apoplastic $[\text{Ca}^{2+}]$ can be 5-1000-fold higher⁵. With the presence of excess $[\text{Ca}^{2+}]$, the XCP1 CNYDase activity can be significantly induced if the pH is increased from normal apoplastic value (5.5) to alkalization apoplastic value (6.0). This may suggest that the XCP1 activity is highly suppressed in the cytosol and the stress-induced apoplast alkalization could be a factor that triggers the activity of XCP1 to process CNYD in PR1. Together with the work in the tomato, we cited in the discussion section, it can be suggested that the XCP1 processes the PR1 in the apoplast region since PR1 and XCP1 are both highly conserved in land plants. In the discussion section, we added a few sentences to address this outstanding question (Line 297-309 in the revised manuscript).

Comment 2-2

In addition to the timing of the AtPR1 processing by XCP1, my other question is whether AtCAPE9 itself move systemically to induce systemic acquired resistance (SAR), or does AtCAPE9 activate SA signalling pathway at local tissue to induce SAR? The authors could clarify the mechanism by checking CAPE9-induced resistance against Pst DC3000 in SA-biosynthesis, SA-signalling, or SAR mutants.

Response:

The reviewer raises an interesting question regarding whether AtCAPE9 induces systemic acquired resistance (SAR) by moving systemically or activating the SA signaling pathway at the local tissue level. As suggested by the reviewer, we have compared the flg22- and AtCAPE9-triggered SAR against *Pst* DC3000 infection in WT, SA-biosynthesis mutant *ics1* and SA-perception mutant *npr1* (Supplementary Fig. 15 and Line 241-246 in the revised manuscript). The SAR elicited by local flg22 in WT was almost abolished in *ics1* and *npr1*, showing the importance of SA signaling in flg22-triggered SAR. Interestingly, local treatment of AtCAPE9 still can activate SAR not only in WT but also in *ics1* and *npr1*, suggesting AtCAPE9 may function as a mobile or elicit other signals for systemic immunity. So far, investigating this mechanism to answer this outstanding question is beyond the scope of our current paper. To demonstrate systemic movement, we would need to perform grafting experiments using an AtCAPE9-receptor mutant, which is the focus of our ongoing work. We have identified the AtCAPE9-receptor, but our results suggest that the SAR induced by AtCAPE9 is not dependent on SA but requires remote/local perception of AtCAPE9 with cooperation with other mobile signals. We plan to submit a manuscript related to this effort later.

Comment 2-3

The authors found that the 'CYND' motif in AtPR1 was cleaved by XCP1. A previous study by the authors (Chen et al., Plant Cell 2014) indicated that the 'CYND' motif is conserved not only in CAPE9/PR1 but also in CAPE1-7. Do the authors think XCP1 is also involved in the processing of these other cytokines? If not, how could specificity be determined? Does a cape9 mutant phenocopy xcp1, or does one need to mutate additional CAPEs?

Response:

We appreciate the reviewer's question regarding the potential involvement of XCP1 in processing the CYND motif in other CAPE cytokines. While it is possible that XCP1 may also process other CAPE members, specificity for proteolytic production may be determined by various factors such as expression levels, localization, and post-translational modifications. We cannot exclude the possibility of XCP1 processing other CAPE members. Regarding our focus on AtCAPE9/PR1 in SAR, we performed peptidome profiling in SA-treated plants and identified AtCAPE9 as the only CAPE member in *Arabidopsis* leaves. We have included this result in the manuscript (Supplementary Data 2 and Supplementary Fig. 1) and used it to address the mutation of AtCAPE9/PR1 can abolish SAR induction (as shown in Fig. 2) (Line 331-334 in the revised manuscript). The function of other CAPE peptides may be similar to CAPE1, its precursor gene is not mainly expressed in leaves or the peptide is not highly produced in the SAR responses.

Comment 2-4

The authors found that SA or INA treatment enhanced the CNYDase activity of XCP1, and the AtCAPE9 was released. On the other hand, treatment with AtCAPE9 enhanced SA content and induced SAR against Pst DC3000. During the infection by Pst DC3000, which process happens first? Does infection of Pst DC3000 enhances the CNYDase activity of XCP1 and activate the production of CAPE9 and subsequent SA accumulation? Or does infection of Pst DC3000 induces SA accumulation, which then enhances the CNYDase activity of XCP1 and production of AtCAPE9, leading to much higher SA accumulation?

Response:

To the best of our knowledge, the first defense triggered by *Pst* DC3000 is related to the PTI elicited by PAMP flg22⁶. Flg22 can soon trigger SA directly⁷ and SA is perceived by NPR1 for largely inducing PR1 expression⁸. The overexpressed PR1 can then facilitate XCP1 to produce more AtCAPE9. The CNYDase activity is more complicated since it depends on how SA regulates XCP1 protein expression and the dynamics of apoplastic pH. So far, we do not have a good method to trace apoplast pH and XCP1 expression in the leave simultaneously. Our data only can suggest XCP1 is involved in the amplification of SA production once SA is first induced by PAMP. To clarify this, we added a suggested

pathway regarding the signaling of AtCAPE9 and other signals (Fig. 7 and Line 317-320 in the revised manuscript).

Comment 2-5

The characterisation of AtPR1 variants, particularly in the context of SAR induction (Fig. 2) would be preferable in stable transgenic lines, rather than after Agrobacterium-mediated transient expression.

Response:

We understand the reviewer's point. The stable line is preferable for complementation since this can provide a stable gene expression for biological assays. However, we used the *Agrobacterium*-mediated transient expression not only to complement *PR1* back to the *pr1* but also to limit the expression of native or modified *PR1* sequences only in the local leaves. In this study, we observed significant induction of SAR when native PR1 was present in the local leaves of *pr1* by transient expression. Moreover, the variation of SAR induction with the presence of native PR1 (P19+PR1) was smaller than in other treatment groups. As for protein and gene expression levels are the primary concern when using a transient expression system, we have provided the data for confirming the transient gene/protein expression each time we performed the experiments (Supplementary Fig. 2d, e).

Comment 2-6

From my understanding the infection assays in Fig. 6 are done with individual F2 plants – this is not sufficient to draw conclusions. These assays need to be repeated at least 3 times with homozygous T3 plants.

Response:

We thank the reviewer for highlighting our mistake in describing the plants used for the infection assays. We apologize for any confusion that may have caused. To clarify, the plants used for biological replicates in each assay were all the progenies of F2 plants. The F2 plants were generated from the crossing of WT with T3 homozygote *xcp1*. Each line obtained in F2 and the F3 lines selected for infection assay were also genotyped. We have added the genotyping result of the F3 plants we used in Fig. 6 and employed a new nomenclature for the lines described in this work as suggested by the reviewer (Fig. 6 and Supplementary Fig. 13 in the revised manuscript).

Comment 2-7

It is also puzzling that infection assays with *Pst* DC3000 are scored at 5 dpi or even 7 dpi, which is certainly not standard in the field. Also, why using seedlings in Fig. 1a but leaves from adult plants

in other figures for such assays?

Response:

We observed the most significant difference in disease symptoms between each leaf sample at 5 to 7 dpi in these *Pst* DC3000-inoculation assays (Fig. 2bc, Fig.6b and 6f, and Supplementary Fig. 15). For tracking the disease symptoms and bacterial population at the same time point, after taking pictures of these leaf samples for disease severity, these leaf samples were soon ground and plated on the media for the measurement of bacterial colony. As shown, the plant resistance represented by disease severity showed a consistent trend with the results represented by bacterial CFU (Fig. 2b, c).

Regarding using seedlings in Fig. 1a, this is because the local anti-pathogen activity induced by AtCAPE9 has been demonstrated using the leaves from adult plants in our previous work⁹, but not yet in the seedling. We have included a sentence to mention the previous result we obtained from adult plants in the revised manuscript (Line 79-80 in the revised manuscript).

Comment 2-8

Regarding the cleavage assays, 10 h seems like a very long incubation time (e.g. Figs. 3 and 4). There also seems to be considerable variation between experiments in terms of RFU. Please comment.

Response:

In the CNYDase activity test, although 5h incubation time is more commonly used for evaluating protease activity¹⁰, there was no significant difference in RFU variation between 5 h and 10 h incubation time (as shown in the figure below). The absolute RFU value depends on the XCP1 expression/modification (for measuring the protease activity total protein lysate), soil composition, soil microbiota (may relate to induced systemic resistance, ISR) or metal ion levels in each batch of the collected plants. Those factors contribute more to the variation of RFU. We measured CNYDase substrate RFU at a specific time point of the individual plants collected from the same batch. This measurement is only for comparing the factors affecting CNYDase activity. In determining the kinetic value of purified XCP1, we use catalytic velocity V_0 of in RFU/ $\mu\text{g} \cdot \text{min}$ to obtain a referenceable value.

Figure. Time course analysis of the fluorescence signal (in RFU) from fluorophore produced by XCP1-His with Ac-CNYD-AMC substrate in different concentrations.

Comment 2-9

Is XCP1 also required for SAR induced by an ETI-inducing PST DC3000 strain (e.g. *AvrRpm1*), as most commonly used in the study of SAR in *Arabidopsis*?

Response:

We thank the reviewer for this insightful question. As mentioned in the manuscript, the flg22 peptide from the bacterial flagellum is a conserved elicitor of PTI in different plant species, which is the first line of plant defense response. Various pathogens, including avirulent and virulent strains, trigger PTI, leading to basal resistance response. Our study reveals that XCP1-mediated cleavage of PR1 is critical for flg22-induced SAR, and thus, it is reasonable to speculate that XCP1 is also required for SAR induced by an ETI-inducing *Pst* DC3000 strain. This is because plants generally perceive flg22 from *Pst* DC3000 before the perception of the bacterial effectors. We have included a hypothesized mechanism based on the discovery of this work in the revised manuscript (Fig. 7 and Line 317-320 in the revised manuscript).

Comment 2-10

Please retain the name XCP1 in the text, as it is confusing to rename genes/proteins that were already named before. Please only refer to the protease as ESCAPE before it is identified.

Response:

We thank the advice from the reviewer. We agree that renaming genes/proteins could cause confusion, and we have revised the manuscript to retain the original name XCP1 throughout the text. We have only referred to the term "ESCAPE" when discussing its identification process and role within the plant.

Minor Issue/Point 2-1

Lines 35, 56, 287: “PR1” should be “AtPR1”.

Response:

We thank the reviewer for identifying the inconsistent naming of PR1. According to the suggestion from reviewer 1, since this is all *Arabidopsis* work, all the AtPR1 terms in the manuscript have been modified to PR1.

Minor Issue/Point 2-2

Line 67: In the final paragraph of the Introduction, the authors should write the main claim/message of the manuscript.

Response:

We thank the reviewer for reminding us of this critical point. We have replaced the last sentence in the final paragraph of the introduction with the main claim of this study (Line 67-72 in the revised manuscript).

Minor Issue/Point 2-3

Lines 141-150: the explanation of how the authors narrowed down the list of Arabidopsis papain-like cysteine protease candidates is unclear. The authors claim that they chose the protease showing 1) 35-kDa molecular weight and 2) similar expression patterns with AtPR1. It is hard to choose the target proteases based on the molecular weight by immunoblotting because the MWs all the proteases listed by authors only differ by a few kDa. Furthermore, the expression pattern of XCP2 (AT1G20850), At3g49340, XBCP3 (At1g09850), At3g19400, At4g19390, At2g27420 and CEP3 (AT3G48350) are similar to XCP1 and AtPR1. However, the authors said XCP1 is the only candidate that fits the criteria. Did the authors have any other evidence that XCP1 is the candidate protease? Did the authors perform any functional screening of the proteases to narrow down the candidates?

Response:

We agree with the reviewer that using a few kDa as the window to rule out those candidates with expression patterns is not reasonable, especially XCP2. We have revised the paragraph in Line 141-150 of the original manuscript to address the concern of our preliminary screening. We have taken the suggestion from reviewer 1 by including 9 PLCP mutants as additional screening evidence (Line 166-168 in the revised manuscript). The abolished biotin-CNYD-CHO binding in *xcp1* (shown in Fig. 5a) can further explain that the XCP1 is the major ESCAPE and CNYDase in the plant (Line 169-171 in the revised manuscript).

Minor Issue/Point 2-4

Lines 144-145: Also, why mentioning MCs initially as focus of investigation as they anyway do not fullfil the criteria?

Response:

We mentioned plant metacaspases as they are well-known caspase-like proteases in plants and first identified a novel cytokine substrate domain that exhibited distinct properties from the substrate of known plant metacaspases. In addition to the MC, the PLCP family also can activate the cytokine for plant immunity.

Minor Issue/Point 2-5

Line 191, 863: “anti-PR1 antibody” should be “anti-AtPR1 antibody”.

Response:

We thank the reviewer for identifying the inconsistency of the naming. As suggested by reviewer 1, we have changed all the AtPR1 into PR1.

Minor Issue/Point 2-6

Fig. 2c, 3a, 4b, 5c, S6: Additional label is written behind the label. Please remove it.

Response:

This issue was due to the conversion of the original file to PDF format. We have removed additional labels after PDF conversion in Fig. 2c, 3a, 4b, 5c and S6 in the original manuscript by reformatting the labeling in those figures (Fig. 2c, 3a, 4b, 5c and Supplementary Fig. 6b in the revised manuscript).

Minor Issue/Point 2-7

Fig.3b, 6c, S10: The vertical axis is not an appropriate scale. Please revise it

Response:

The scales for those figures have been revised (Fig. 3b, 6c and Supplementary Fig. 11c in the revised manuscript).

Minor Issue/Point 2-8

Fig.6a, 6b, 6e, 6f, S1a, S1d, S3, S5, S7c: Line frames are displayed because of the backfill.

Response:

The line frame issues in Fig.6a, 6b, 6e, 6f, S1a, S1d, S3, S5, S7c has been fixed (Fig.6a, 6b, 6e, 6f; Supplementary Fig. 2a, 2d, 4, 6, 7c in the revised manuscript).

Minor Issue/Point 2-9

Fig.S5d: The authors should perform statistical analysis for the data.

Response:

The statistical test in Fig. S5d in the original manuscript has been performed (Supplementary Fig. 6e in the revised manuscript).

Minor Issue/Point 2-10

Please could the authors quantify the correlation between the spectra in Fig. 1b.

Response:

We thank the suggestion from the reviewer. We have added the Pearson correlation coefficient to evaluate the similarity of the spectra illustrated in Fig. 1b. Moreover, using the MS/MS ion search (MASCOT) against Arabidopsis hypothetical peptide database to show the matching result endogenous AtCAPE9 in peptidome analysis has been included in Supplementary Fig. 1 (Line 82-87 in the revised manuscript).

Minor Issue/Point 2-11

Please plot bar charts so that individual datapoints are visible.

Response:

Each measured data point for all the bar graphs has been uploaded as Source Data.

Minor Issue/Point 2-12

What do the authors propose the middle band is in Fig. S2 (and other later blots)? Why does this not appear in Fig. 1c? Please could the authors treat the plants in Fig. 1c with SA/INA to see whether this product is still formed under these conditions.

Response:

The production of the middle band in treated plants is not quite stable in Mock or SA/INA treated plants (Fig. S2 in the original manuscript and Supplementary Fig. 3 in the revised manuscript). This band has been reported in another study¹¹. However, we have not been able to determine the cause of this band. It is possible that it represents the processing of PR1-eYFP by another enzyme or post-translational modifications on AtCAPE9-eYFP.

Minor Issue/Point 2-13

Please define Ac-CNYD-AMC the first time it is used.

Response:

The definition of Ac-CNYD-AMC has been added (Line 125-127 in the revised manuscript).

Minor Issue/Point 2-14

Do the authors have a hypothesis as to why mutating most of the cysteine proteases in S6 leads to significant enhancement of cleavage of the CNYD reporter?

Response:

We are currently unable to have a specific hypothesis for this good question. This may be due to the induction of expression of XCP1 or some other novel peptidases when a specific protease is mutated. The absence of a specific enzyme may significantly alter the protein interaction network, affecting not only the expression but also the pH, Ca²⁺, and posttranslational modification of XCP1 for the enzyme activity. It has been reported that the loss of some protease's expression could induce the expression or activity of other proteases^{12, 13}.

Minor Issue/Point 2-15

Could the endogenous CAPE9 levels be normalized to an endogenous peptide from a housekeeping protein?

Response:

The reviewer raised a critical issue regarding the normalization of endogenous peptide levels. We agree that normalizing to an endogenous peptide from a housekeeping protein could be useful. However, as of now, there is no standardized and reliable method for normalizing endogenous peptide levels. To address this, we employed an integrated normalization strategy by doping a synthetic peptide or tryptic peptides from a standard protein to the peptide extract and saturating the SPE cartridge with a fixed loading capacity by the peptide extract.

In order to identify an appropriate endogenous peptide as a normalization reference, it would be necessary to collect a large set of peptidome analysis results. Such an analysis would allow us to determine which endogenous peptide can serve as a reliable internal standard. We appreciate the reviewer's suggestion and will consider this approach in future studies.

Minor Issue/Point 2-16

Fig. 5b should have a loading control

Response:

We think the reviewer should notice there is a missing loading control in Fig. 5a. We have included the loading control in Fig. 5a. In Fig. 5b, we did the protein quantitation before testing the activity.

Minor Issue/Point 2-17

Please change the nomenclature for the crosses – currently it makes me think the lines are F1s

rather than being homozygous.

Response:

We thank the reviewer identified the nomenclature issue in this paper. All of the cross lines we showed were at least F2. All the plants used in enzyme activity assay and pathogen assay were F3 generation. We have changed the nomenclature for the cross lines used in this manuscript.

Minor Issue/Point 2-18

Refs 11 and 13 are the same.

Response:

The redundant reference has been removed.

Minor Issue/Point 2-19

Throughout the manuscript: as all forms of plant immunity are innate, it does not make sense to refer to plant innate immunity.

Response:

All of the “innate immunity” has been changed to “immunity”.

Minor Issue/Point 2-20

Line 244: precise that these findings were in tomato.

Response:

The sentence to address the finding was in other plants has been added (Line 272-273 in the revised manuscript).

References (from reviewers):

- Baena, G., Xia, L., Waghmare, S., & Karnik, R. (2022). SNARE SYP132 mediates divergent traffic of plasma membrane H⁺-ATPase AHA1 and antimicrobial PR1 during bacterial pathogenesis. *Plant physiology*, 189(3), 1639–1661. <https://doi.org/10.1093/plphys/kiac149>
- Pečenková, T., Pejchar, P., Moravec, T., Drs, M., Haluška, S., Šantrůček, J., Potocká, A., Žárský, V., & Potocký, M. (2022). Immunity functions of Arabidopsis pathogenesis-related 1 are coupled but not confined to its C-terminus processing and trafficking. *Molecular Plant Pathology*, 23(5), 664–678. <https://doi.org/10.1111/mpp.13187>
- Chen, Y. L., Lee, C. Y., Cheng, K. T., Chang, W. H., Huang, R. N., Nam, H. G., & Chen, Y. R. (2014). Quantitative peptidomics study reveals that a wound-induced peptide from PR-1 regulates immune signaling in tomato. *The Plant Cell*, 26(10), 4135–4148. <https://doi.org/10.1105/tpc.114.131185>

Reference cited by authors

1. Rotonda J, *et al.* The three-dimensional structure of apopain/CPP32, a key mediator of apoptosis. *Nat Struct Biol* 3, 619-625 (1996).

2. Lincoln JE, Sanchez JP, Zumstein K, Gilchrist DG. Plant and animal PR1 family members inhibit programmed cell death and suppress bacterial pathogens in plant tissues. *Mol Plant Pathol* **19**, 2111-2123 (2018).
3. Kong XP, *et al.* Antagonistic Interaction between Auxin and SA Signaling Pathways Regulates Bacterial Infection through Lateral Root in Arabidopsis. *Cell Reports* **32**, (2020).
4. Vercammen D, *et al.* Type II metacaspases Atmc4 and Atmc9 of Arabidopsis thaliana cleave substrates after arginine and lysine. *Journal of Biological Chemistry* **279**, 45329-45336 (2004).
5. Kader MA, Lindberg S. Cytosolic calcium and pH signaling in plants under salinity stress. *Plant Signal Behav* **5**, 233-238 (2010).
6. Chang M, Chen H, Liu FQ, Fu ZQ. PTI and ETI: convergent pathways with diverse elicitors. *Trends in Plant Science* **27**, 113-115 (2022).
7. Tsuda K, Sato M, Glazebrook J, Cohen JD, Katagiri F. Interplay between MAMP-triggered and SA-mediated defense responses. *Plant J* **53**, 763-775 (2008).
8. Rochon A, Boyle P, Wignes T, Fobert PR, Despres C. The coactivator function of Arabidopsis NPR1 requires the core of its BTB/POZ domain and the oxidation of C-terminal cysteines. *Plant Cell* **18**, 3670-3685 (2006).
9. Chen YL, *et al.* Quantitative peptidomics study reveals that a wound-induced peptide from PR-1 regulates immune signaling in tomato. *Plant Cell* **26**, 4135-4148 (2014).
10. Wilkins KA, Bancroft J, Bosch M, Ings J, Smirnoff N, Franklin-Tong VE. Reactive Oxygen Species and Nitric Oxide Mediate Actin Reorganization and Programmed Cell Death in the Self-Incompatibility Response of Papaver. *Plant Physiology* **156**, 404-416 (2011).
11. Pecenkova T, *et al.* Immunity functions of Arabidopsis pathogenesis-related 1 are coupled but not confined to its C-terminus processing and trafficking. *Mol Plant Pathol* **23**, 664-678 (2022).
12. Gruis DF, Selinger DA, Curran JM, Jung R. Redundant proteolytic mechanisms process seed storage proteins in the absence of seed-type members of the vacuolar processing enzyme

family of cysteine proteases. *Plant Cell* **14**, 2863-2882 (2002).

13. Bollhoner B, Prestele J, Tuominen H. Xylem cell death: emerging understanding of regulation and function. *J Exp Bot* **63**, 1081-1094 (2012).

Reviewer #1 (Remarks to the Author):

The issues raised have been sufficiently addressed by the authors. I like to congratulate the authors with this really great discovery!

Reviewer #2 (Remarks to the Author):

I think the authors have responded well to the reviewers comments. There are still some issues regarding the use of uncommon methods (regarding pathogen assays using seedlings and longer DPI, and enzymatic activity tests with more prolonged incubation than usual). However, they have controls for these experiments, and their answers to reviewers are reasonable.